# MEIcoder: Decoding Visual Stimuli from Neural Activity by Leveraging Most Exciting Inputs

**Jan Sobotka**[1,*]     **Luca Baroni**[2]     **Ján Antolík**[2]

[1]EPFL, Switzerland   [2]Charles University, Czechia

## Abstract

Decoding visual stimuli from neural population activity is crucial for understanding the brain and for applications in brain-machine interfaces. However, such biological data is often scarce, particularly in primates or humans, where high-throughput recording techniques, such as two-photon imaging, remain challenging or impossible to apply. This, in turn, poses a challenge for deep learning decoding techniques. To overcome this, we introduce MEIcoder, a biologically informed decoding method that leverages neuron-specific most exciting inputs (MEIs), a structural similarity index measure loss, and adversarial training. MEIcoder achieves state-of-the-art performance in reconstructing visual stimuli from single-cell activity in primary visual cortex (V1), especially excelling on small datasets with fewer recorded neurons. Using ablation studies, we demonstrate that MEIs are the main drivers of the performance, and in scaling experiments, we show that MEIcoder can reconstruct high-fidelity natural-looking images from as few as 1,000-2,500 neurons and less than 1,000 training data points. We also propose a unified benchmark with over 160,000 samples to foster future research. Our results demonstrate the feasibility of reliable decoding in early visual system and provide practical insights for neuroscience and neuroengineering applications.

## 1   Introduction

Recent progress in machine learning (ML), together with advances in collecting single-cell brain activity data, has enabled powerful data-driven approaches to model the brain. The dominant approach is to use ML models to characterize the stimulus-response function, i.e., to predict brain activity in response to external variables (*encoding*), such as visual stimuli [2, 9, 24, 27, 28, 46]. The inverse problem of *decoding* high-fidelity stimuli from brain activity started to garner attention only relatively recently [6, 8, 18, 23, 33, 37]. One of the main reasons for this is the inherent difficulty of decoding high-information content, such as images, from a small number of neurons that provide a highly compressed and noisy version of the original stimulus [21]. This inverse problem is exaggerated by the scarcity of single-subject data, which is necessary to accurately capture the unique response-stimulus mapping of the subject's visual system. Therefore, there is a need for decoding techniques that can learn from limited training examples and from a small number of recorded neurons.

However, training machine learning models from scratch on such scarce single-subject data tends to yield low-fidelity image reconstructions that lack sufficient detail [18, 33, 51]. Conversely, leveraging pre-trained models, particularly those from the field of Generative AI (GenAI), provides high-resolution but unreliable reconstructions, often containing hallucinated content [31, 35, 37]. Indeed, as [39] and our results in subsection 4.5 show, prior decoding methods employing (text-guided) GenAI are heavily biased toward their pre-training distribution of semantically rich images. This leads to deceptively realistic, yet often pixel-level inaccurate, image reconstructions that do not capture

---

*Correspondence to: `jan.sobotka@epfl.ch`

39th Conference on Neural Information Processing Systems (NeurIPS 2025).

the true fine spatial characteristics of the visual stimulus (and occasionally outright hallucinations), hindering the reliability of these methods and limiting their scope of application [21].

To remedy these problems, we develop an end-to-end trained decoding method called MEIcoder, which utilizes prior knowledge from computational neuroscience and adversarial objectives to outperform previous methods on three difficult datasets. MEIcoder achieves high-fidelity reconstructions by (1) utilizing a strong computational prior in the form of neuron-specific *most exciting inputs* (MEIs), (2) a novel training loss based on the structural similarity index measure (SSIM), (3) an auxiliary adversarial training objective that pushes reconstructions toward a manifold of natural-looking images, and (4) a parameter-efficient architecture that allows training on multiple distinct datasets. In summary, the main contributions of this paper are as follows:

1. We develop a method that achieves state-of-the-art performance in decoding visual stimuli from neural population activity in V1. This result demonstrates that decoding high-fidelity reconstructions is *feasible* with the currently available single-subject data.

2. To understand the scaling behavior of our model, we analyze the relationship between performance, the number of available recorded neurons, and the amount of training data.

3. To stimulate further developments in this area, we aggregate datasets from multiple sources into a decoding benchmark with over 160,000 samples.

## 2   Related work

Most previous work on decoding brain signals has been done with magnetoencephalography (MEG) and functional magnetic resonance imaging (fMRI) data. For example, [6] leveraged pre-trained image embeddings and a pre-trained image generator to perform real-time decoding of MEG signals into images. [30, 35, 36, 37, 41] used GenAI techniques, such as pre-trained diffusion models [19, 40], to decode images from fMRI. Their techniques were able to reconstruct the semantic information in the visual stimuli, such as object categories, but were unable to capture low-level features of the images. Furthermore, a study by [39] provided formal analysis to demonstrate that prior decoding approaches based on diffusion models suffer from so-called "output dimension collapse", which restricts their decodable features. Their study, as well as our results in subsection 4.4, also show that prior GenAI-based decoding techniques tend to hallucinate, leading to untrustworthy and spatially inaccurate reconstruction of novel images. These findings highlight the importance of choosing an appropriate prior and carefully balancing it with the neural data to achieve reliable reconstructions.

One of the first studies investigating decoding from neuron-level data leveraged known retinotopy to reconstruct simple visual stimuli and mental imagery [43]. Later, using Generative Adversarial Networks (GANs) [17] and other deep learning approaches, a series of works [18, 25, 31, 33, 51] showed promising initial results in decoding more complex stimuli from higher-order areas of the visual system, such as V4 and the inferior temporal cortex. For example, [25] incorporated known biological properties of neurons in the visual system into their brain-inspired architecture to reconstruct images from sequences of spikes. More recently, [23] introduced a homeomorphic decoder with learned inverse retinotopic mapping to reconstruct naturalistic images from macaque brain signals. As we evaluate their method in subsection 4.4, we find that its fully end-to-end trained retinal embeddings are incapable of reconstructing high-fidelity images from our limited mouse data. Moreover, their architecture is not designed to work with multiple distinct datasets, which prevents it from integrating learning signals from data across different subjects.

Instead of training to decode images directly, the novel approach from [11] pre-trains a CNN encoder and, at inference time, performs an iterative encoder inversion procedure. It begins by (randomly) initializing the pixels of the reconstructed image and then takes gradient steps on the image pixels to minimize the difference between the ground-truth responses and the responses predicted by the encoder from the reconstructed image. In addition to the computational burden at inference time, this approach does not directly optimize for reconstruction quality at the pixel level, but rather for reconstruction in the space of neuronal responses predicted by the encoding model. This can lead to image artifacts and potentially limit its ability to accurately reconstruct target images.

Lastly, similarly as for the fMRI data, [31] leveraged a pre-trained diffusion model and a CNN encoder to decode neural population activity. More specifically, their method, *Energy Guided Diffusion*, guided the inversion of a CNN encoder using a frozen diffusion model. This approach enabled sharper

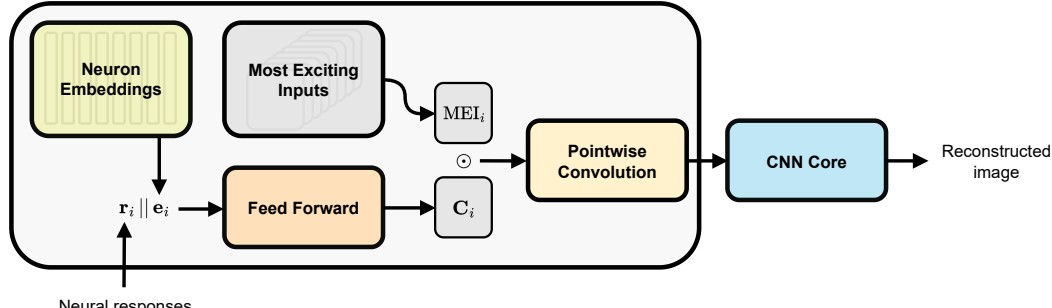

Figure 1: Architecture of MEIcoder. First, individual neural responses $\mathbf{r}_i$ and their corresponding neuron embeddings $\mathbf{e}_i$ get projected by a feed-forward network into context representations $\mathbf{C}_i$ (left). The context representations are then pointwise-multiplied with MEIs and passed through a pointwise convolution layer (center). Finally, the output of the convolution layer is used by the CNN core to reconstruct the final image (right).

reconstructions by introducing a strong bias toward the image statistics of the pre-trained diffusion model, but still optimized for reconstruction quality in the neural activity space as captured by the encoding model, rather than in the image pixel space. In turn, the resulting reconstructions from neuronal responses often contained spurious features that were not present in the original images, which we also confirm in our experiments (subsection 4.4). Overall, these challenges highlight the need for a new decoding approach that mitigates hallucination, preserves fine-grained visual details, and more effectively balances neural evidence with generative priors.

## 3 MEIcoder

To overcome the limitations of previous methods, we introduce MEIcoder. It consists of two components: a single *core* module and one or more *readin* modules. Each readin is trained separately for its respective single-subject dataset and acts as an embedding function of the neural activity into the core's latent space. The core, on the other hand, is shared across all possibly heterogeneous multi-subject datasets and maps from its latent space into images. The purpose is to reuse learning signals across recordings from different subjects, even when their number of neurons and response-stimulus mappings differ.

**Core.** To keep MEIcoder parameter-efficient and suitable for low-data regimes, we build the core as a six-layer convolutional neural network (CNN) with batch normalization, ReLU activation function, and dropout. Further details and hyperparameters can be found in subsection A.3.

**Readin.** One of our main contributions comes in the readin module. While prior work in this area used simple feed-forward neural networks to translate the raw brain signals into the latent space [35, 36], we found it to be highly suboptimal as it quickly overfits and does not generalize well. This motivated us to introduce a novel readin module, where we inject additional prior knowledge and regularization into the decoder in the form of *most exciting inputs* (MEIs). In the context of visual processing, MEIs are images that maximize the response of a given neuron in the visual system, thus being highly informative about the coding properties of individual neurons [4, 32, 45]. They are easily obtainable from the training data, and their generation needs to be done only once (subsection A.4).

The intuition behind MEIcoder stems from the fact that MEIs contain information about the receptive fields of neurons[2]–the spatial patterns that most strongly stimulate them. This suggests a straightforward linear decoding approach: overlay the MEIs of all neurons on top of each other, weighted by their respective neural responses, to reconstruct the image. This intuition is the primary building block of our method. However, in our method, since coding in V1 is not linear, we leave the combination of MEIs onto the nonlinear blocks in the MEIcoder's core. Furthermore, to provide greater flexibility to the decoding process, we add learnable neuron embeddings that can encode additional properties of the neural code. We show the MEIcoder pipeline in Figure 1 and describe it more formally below.

---

[2]In case of linear neurons, MEIs are equivalent to the receptive fields.

Let $\mathbf{r} \in \mathbb{R}^n$ be the vector of responses of $n$ neurons, and $h, w$ be the height and width of the images to decode. The first step of our readin is to independently embed individual neural responses $\mathbf{r}_i$ together with their corresponding learnable neuron embeddings $\mathbf{e}_i \in \{\mathbf{e}_j \in \mathbb{R}^d\}_{j=1}^n$ using a one-layer neural network $g_\psi : \mathbb{R}^{1+d} \rightarrow \mathbb{R}^{h \cdot w}$ into *context representations* $\mathbf{C} \in \mathbb{R}^{n, h \cdot w}$. The second step is to pointwise-multiply the precomputed MEIs $\mathbf{M} \in \mathbb{R}^{n, h, w}$ with the reshaped context representations $\mathbf{C} \in \mathbb{R}^{n, h, w}$ to obtain *neural maps* $\mathbf{H} = \mathbf{M} \odot \mathbf{C}$. Lastly, to obtain a constant number of output channels for readins operating with possibly different numbers of neurons, we apply a pointwise convolution to transform the neural maps of shape $n \times h \times w$ into compressed neural maps $\mathbf{H}_c \in \mathbb{R}^{d_c, h, w}$ which form the input to the core module of the decoder.

## 3.1 Training

Unless stated otherwise, we train the decoder end-to-end from random initialization. The full training objective consists of two terms: (1) SSIM-based reconstruction loss, and (2) adversarial loss. We combine the two using weighting coefficients $\lambda_{\text{SSIM}} = 0.9$ and $\lambda_{\text{ADV}} = 0.1$.

**SSIM-based reconstruction loss.** Given that the images for reconstruction are encoded by natural vision, we employ a modification of the Structural Similarity Index Measure (SSIM) [47] to steer the decoder toward perceptually important image features. Specifically, we use the negative log-SSIM loss defined as:

$$\mathcal{L}_{\text{SSIM}}(\mathbf{y}, \hat{\mathbf{y}}) = -\log\left(\frac{\text{SSIM}(\mathbf{y}, \hat{\mathbf{y}}) + 1}{2} + \epsilon\right), \tag{1}$$

where $\mathbf{y}, \hat{\mathbf{y}} \in \mathbb{R}^{c, h, w}$ are the ground-truth and reconstructed image, respectively, and $\epsilon = 10^{-6}$ is introduced for numerical stability. As later shown in subsection 4.5 with ablation studies, we found this training objective more effective at producing perceptually accurate reconstructions compared to standard objectives such as the mean squared error (MSE). Unlike perceptual loss functions based on embeddings from pre-trained models, which we found to be unstable in training and produced high-frequency artifacts, the SSIM objective required no further tuning and led to consistent results.

**Adversarial training.** The limited amount of data may not give the decoder enough training signal to learn to reconstruct high-fidelity natural-looking images, and may potentially lead to overfitting. To counteract this, we use an auxiliary adversarial objective similar to that used in GANs[3]. More specifically, we train a secondary CNN to classify whether a given input image is a reconstruction from our decoder or a reference (ground-truth) image from the dataset (see subsection A.3 for details). Given this discriminator $\text{D}_\phi : \mathbb{R}^{c, h, w} \rightarrow [0, 1]$, we add the following loss for training the decoder:

$$\mathcal{L}_{\text{ADV}}(\hat{\mathbf{y}}) = \left(\text{D}_\phi(\hat{\mathbf{y}}) - 1\right)^2. \tag{2}$$

Note that unlike standard GANs, our decoder is not trained generatively and is conditioned only on neuronal responses. Moreover, its objective directly optimizes for spatially accurate reconstruction, and its priors are more aligned with the biological vision through the MEIs. These factors make MEIcoder more reliable and faithful to the true response-stimulus function that it is trying to capture.

We train MEIcoder for 300 epochs using the AdamW optimizer [26] with a learning rate and weight decay found using hyperparameter search and the validation dataset. Similar to early stopping [29], we pick the best model from training based on the Alex(5) score measured on the validation dataset.

## 4 Experiments

We compare MEIcoder to state-of-the-art baselines on three datasets, two of which represent data- and neuron-constrained settings. Given that there are currently no unified benchmarks for visual decoding from neural population activity with sufficient heterogeneity, we propose our own as an aggregation of previously published data sources.

### 4.1 Data

**Brainreader dataset.** The BRAINREADER[4] data comes from mouse V1 and was originally introduced by [11]. For our experiments, we use data from a single mouse, where recorded spike traces

---

[3]Previous work has also found adversarial objectives effective for reconstructing visual stimuli [18, 23, 38].
[4]This name, not used by the original authors, is chosen to differentiate this dataset.

were aggregated and averaged over a 500 ms time window following the presentation of grayscale images. We divide the dataset into training, validation, and test sets of 4,500, 500, and 100 samples, respectively. Each data point consists of a $36 \times 64$ px grayscale image sampled from ImageNet [12], along with evoked neuronal responses of 8,587 neurons. For one of our experiments, we use data from 8 mice, where the number of recorded neurons varies between the individual mouse datasets. We refer the reader to subsection A.2 for additional details.

**SENSORIUM 2022 dataset.** We repurpose the mouse dataset published by the SENSORIUM 2022 competition [48] for our decoding task. Specifically, the dataset used in our experiments contains time-binned recordings of responses from 8,372 neurons for 4,984 images, which we split into training and validation sets. For final evaluation, we use a testing set of 100 images with corresponding 10-trial averaged neural responses. For one of our experiments, we pre-train on data from 5 mice available in the SENSORIUM 2022 data corpus and fine-tune on a single-mouse data.

**Synthetic cat V1 dataset.** Lastly, since most of the currently publicly available datasets suitable for decoding in V1 are greatly data-constrained and limited to mouse or monkey data, we leverage a highly biologically realistic spiking model of cat V1 from [3] to generate a large synthetic dataset. This spiking model has been extensively validated in a series of studies, demonstrating a wide range of accurately replicated properties of V1 coding [3, 42, 44]. For data generation, we sample 50,250 grayscale images from ImageNet and encode them into responses of 46,875 neurons using the spiking model. We split this additional synthetically generated dataset into training (45,000), validation (5,000), and test (250) sets. Additional details can be found in subsection A.2.

## 4.2 Evaluation metrics

For robust evaluation, we follow previous studies [22, 23, 36] and use (1) SSIM, (2) Pearson correlation between the pixel values of the reference and the reconstructed image, and (3) feature correlation with two-way identification using a pre-trained AlexNet. For the feature correlation, we follow [36] and extract feature representations at the second and fifth layers of ImageNet-pretrained AlexNet to evaluate the two-way identification ability. This identification score refers to the percentage of correct comparisons assessing if the reference image embedding is more similar to the reconstructed image embedding or to a randomly selected image embedding. We use the implementation of the two-way identification from [36]. For additional experimental details, please refer to subsection A.1.

## 4.3 Baselines

We compare MEIcoder with five baseline methods, four of which were developed on population recordings of single-cell activity, and one of which was originally designed on fMRI. The first two baselines are the Inverted Encoder (InvEnc) [11] and Energy Guided Diffusion (EGG) [31] as described in section 2. InvEnc showed state-of-the-art performance on the BRAINREADER dataset, and both methods are representative of a class of decoding methods that invert pre-trained encoder models for image reconstruction, minimizing errors in response space.

The third baseline is the homeomorphic decoder *MonkeySee* [23]. It employs the U-Net architecture, feature representations from a pre-trained CNN, and is trained end-to-end with VGG feature loss, L1 reconstruction loss, and an adversarial objective. Another baseline similar to MonkeySee is the *CAE* decoder [10], which employs a series of fully-connected layers followed by downsampling and upsampling convolutional blocks. Both methods showed improvements over the previous best-performing approaches, and we consider them strong representatives of direct pixel decoding.

The last baseline to which we compare is MindEye2 [36], which combines multiple pre-trained models from the field of GenAI with end-to-end trained networks. It demonstrated high-fidelity reconstruction on the fMRI Natural Scenes Dataset [1], outperforming all previous methods.

For all five baselines, we use the code provided by the original authors. To find the optimal hyperparameters, we use the same procedure as for our method: we perform a hyperparameter search using only the training and validation datasets, and pick the best model checkpoint from training based on the performance on the validation data.

Table 1: Results on the test sets from the BRAINREADER and SENSORIUM 2022 datasets. Best results are highlighted **red** and second-best in **bold**. All values are means over three random seeds; standard deviations and results for higher-level metrics are available in Tables 3 and 9, respectively.

| Method | BRAINREADER | | | | SENSORIUM 2022 | | | |
|---|---|---|---|---|---|---|---|---|
| | SSIM | PixCorr | Alex(2) | Alex(5) | SSIM | PixCorr | Alex(2) | Alex(5) |
| InvEnc | .321 | .611 | .989 | .896 | .288 | .453 | .915 | .720 |
| EGG | .256 | .495 | .758 | .659 | .256 | .365 | .777 | .755 |
| MonkeySee | .232 | .565 | .967 | .826 | .185 | .338 | .564 | .523 |
| CAE | .256 | .638 | .930 | .730 | .287 | **.539** | .656 | .549 |
| MindEye2 | .277 | .560 | .946 | .878 | .210 | .471 | .877 | .762 |
| MindEye2 (FT) | .234 | .516 | .920 | .836 | .243 | .499 | .918 | .799 |
| MEIcoder | **.400** | **.679** | **.998** | **.990** | **.331** | .503 | **.988** | **.896** |
| MEIcoder (FT) | **.424** | **.706** | **.999** | **.977** | **.318** | .486 | **.975** | **.908** |

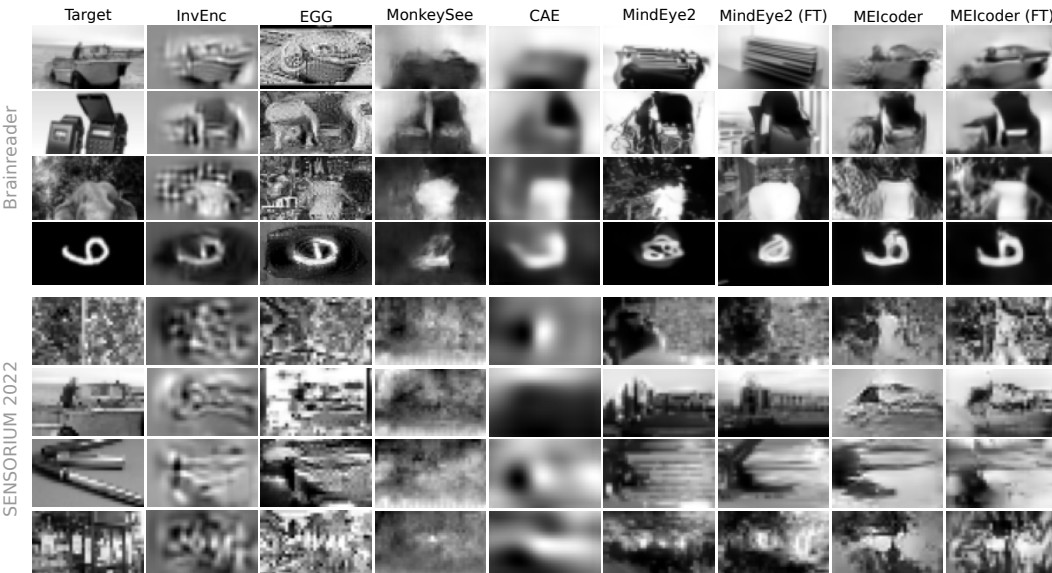

Figure 2: Reconstructions on the BRAINREADER (top) and SENSORIUM 2022 (bottom) datasets.

## 4.4 Results and discussion

Tables 1 and 2 show that MEIcoder outperforms all baselines in decoding visual stimuli on all three datasets (considered as an aggregate over the metrics). Furthermore, we can see that the improvements are most significant in the biological data-scarce and neuron-scarce settings (BRAINREADER and SENSORIUM 2022 datasets), highlighting the suitability of MEIcoder for such cases. Closer qualitative evaluation in Figure 2 further confirms that MEIcoder handles the low-data regime well, producing more detailed and faithful reconstructions compared to other methods. For example, while the MindEye2 reconstructions on the BRAINREADER dataset are relatively sharp but spatially inaccurate, and MonkeySee reconstructions are the opposite, MEIcoder achieves both at the same time. This result can be explained by the difference in prior knowledge injected into these three decoders. Namely, MindEye2 reconstructions are steered toward natural-looking, semantically rich images due to its GenAI components, and MonkeySee employs a weak prior in the form of frozen feature embeddings from a pre-trained CNN. MEIcoder, by contrast, leverages MEIs as a prior that is more closely aligned with the neurons from which it is actually decoding.

We also test how well MEIcoder handles multi-subject[5] pre-training and subsequent single-subject fine-tuning (FT). Comparing the FT version against others in Table 1 and Figure 2, we can see that the split of the architecture into a core and subject-specific readins allows MEIcoder to work well in this more heterogeneous training regime, outperforming the single-subject training in some cases.

---

[5]Combined data from 8 and 5 mice from BRAINREADER and SENSORIUM 2022 datasets, respectively.

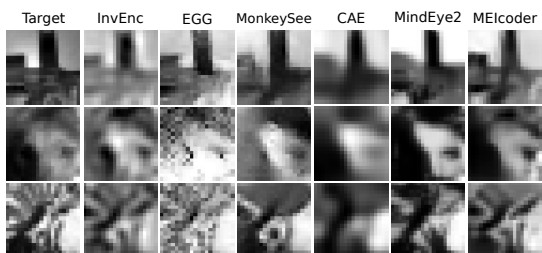

Figure 3: SYNTHETIC CAT V1 reconstructions. See subsection A.12 for more examples.

Table 2: Results on the test set of the SYNTHETIC CAT V1 dataset. Best results are highlighted in red and second-best in **bold**. All values are means over three random seeds; standard deviations are available in Table 4.

| Method | SYNTHETIC CAT V1 | | | |
|---|---|---|---|---|
| | SSIM | PixCorr | Alex(2) | Alex(5) |
| InvEnc | **.771** | .833 | **.986** | **.978** |
| EGG | .640 | .667 | .936 | .906 |
| MonkeySee | .607 | .723 | .982 | .958 |
| CAE | .637 | **.792** | .927 | .776 |
| MindEye2 | .559 | .757 | .977 | .939 |
| MEIcoder | .774 | .777 | .994 | .987 |

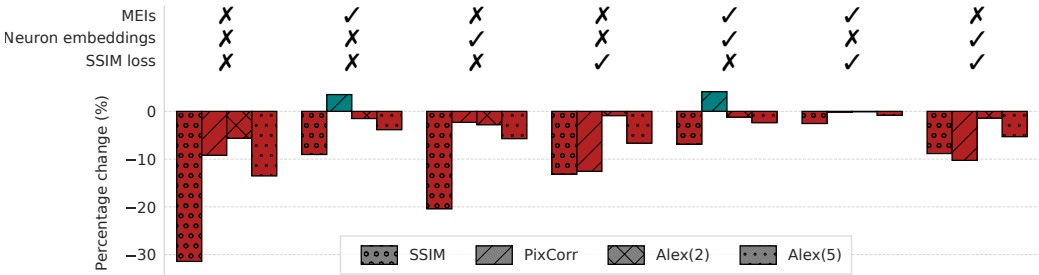

Figure 4: Ablation study on sub-components of MEIcoder. The percentage (y-axis) is calculated with respect to a setting with no ablations. Reported values are the average across the three datasets.

## 4.5 Further analysis

**Ablation study.** We conduct ablation studies to quantify the importance of individual components of MEIcoder. Specifically, we evaluate performance after removing MEIs from the readin module by using only the context representations $\mathbf{C}$ as the neural maps $\mathbf{H}$ (3). Similarly, we evaluate the metrics after removing the neuron embeddings $\mathbf{e}_i$ from the input of the neural network $g_\psi$, and after substituting the standard MSE training objective in place of the SSIM-based reconstruction loss. We report the average over the three datasets.

The results in Figure 4 show that MEIs have the most significant positive influence on the state-of-the-art performance of MEIcoder. Indeed, compared to the individual ablations of the SSIM loss or the neuron embeddings, the removal of MEIs leads to the largest drop in performance in terms of all metrics. This result demonstrates the usefulness of MEIs as prior knowledge in this low-data regime.

**Variance of errors.** Here, we provide further evidence for the shortcomings of methods that rely on pre-trained GenAI models. Specifically, we quantify the reliability of MEIcoder and MindEye2 by examining the variance in their reconstruction quality across individual data points in the test set. As shown in Figure 5, both the Alex(2) and Alex(5) scores fluctuate much more for MindEye2, indicating that MEIcoder provides more consistent results. Figure 6 also shows examples of hallucinated content in MindEye2 images, supporting previous claims about spurious reconstructions [39].

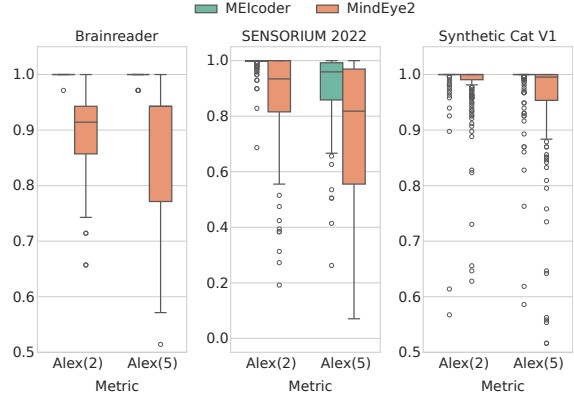

Figure 5: Variance in reconstruction quality.

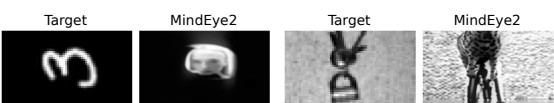

Figure 6: Example of hallucinated content in reconstructed images by MindEye2 (left: woman face; right: giraffe).

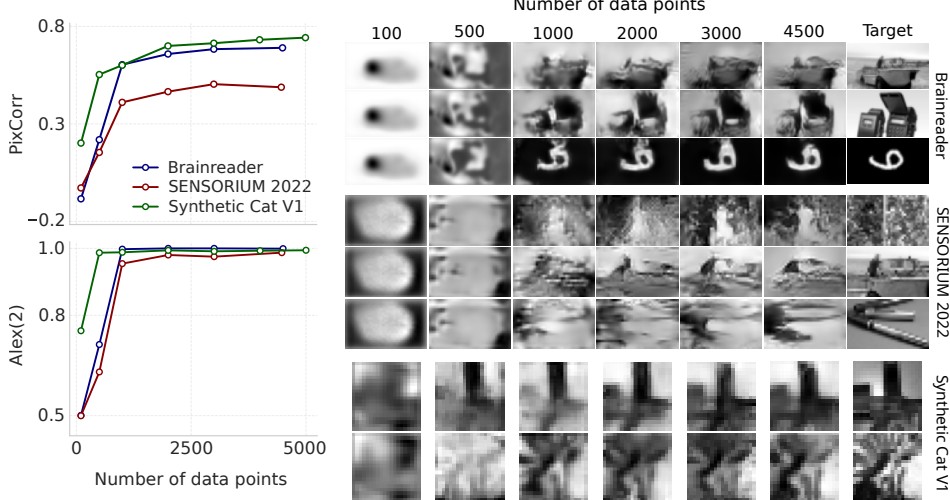

Figure 7: Relationship between MEIcoder's performance and the number of training data points.

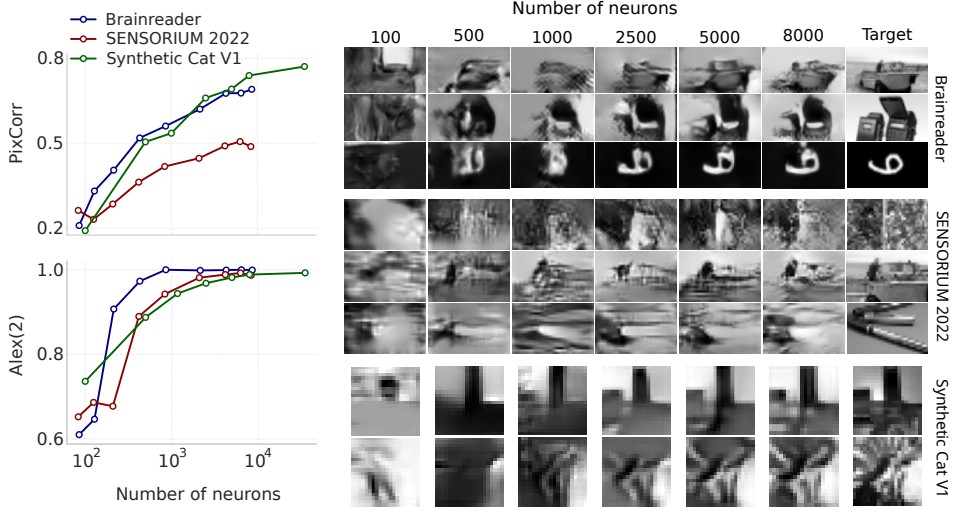

Figure 8: Relationship between MEIcoder's performance and the number of neurons.

**Amount of training data.** To quantify the scaling behavior of MEIcoder, we train it on varying sizes of the training set and evaluate it using the full test set. As we can see in Figure 7, MEIcoder's reconstructions start to capture ground-truth images after training with 1,000 or fewer training data points. Moreover, if we look at the Alex(2) performance of the second-best method from Table 1, which was trained on all data, we see that MEIcoder outperforms it already at 1,000 training data points, demonstrating its data efficiency. Lastly, it is worth noting that while the quantitative measures start to plateau, the qualitative (visual) results keep steadily improving, indicating limitations of the currently widely used metrics, such as PixCorr and two-way identification scores.

**Number of neurons.** Another challenge in reconstructing visual stimuli from neural population activity comes in the form of *information scarcity*, which results from the difficulty of (invasively) recording many cells in parallel and from neuronal noise [13]. This problem manifests when decoding methods need to successfully invert the brain's unique stimulus-encoding process using only a small subset of neurons, which might not carry sufficient information to fully decode the original stimulus. To balance the reconstruction quality with the costs of brain recordings, it is therefore paramount to understand how the decoder's performance scales with the number of neurons. Figure 8 shows this relationship, and we can see that MEIcoder reaches more than 95% two-way identification ability (Alex(2)) on all three datasets with just around 1,000 neurons and can accurately distinguish digits with around 2,500 neurons. Interestingly, the pixel correlation keeps increasing for both BRAINREADER and SYNTHETIC CAT V1 datasets, indicating that the performance does not saturate

even with the 46,875 neurons available in the synthetic cat dataset, which is several-fold more than what most current biological datasets can provide in a single animal.

The scaling experiments above reveal several insights. Firstly, the similar scaling behavior of biological and SYNTHETIC CAT V1 data highlights the potential of high-fidelity spiking models, which capture detailed visual cortex dynamics and provide abundant data for developing and benchmarking decoding methods. Second, the number of available neurons seems to be more limiting than the size of the training dataset, as can be seen by the steady qualitative improvement of reconstructions and unsaturated metrics with an increasing number of available neurons. Finally, our analysis indicates that recording between 1,000 and 2,500 neurons from mouse V1 is enough for fine-grained reconstructions with the power to discriminate between handwritten digits.

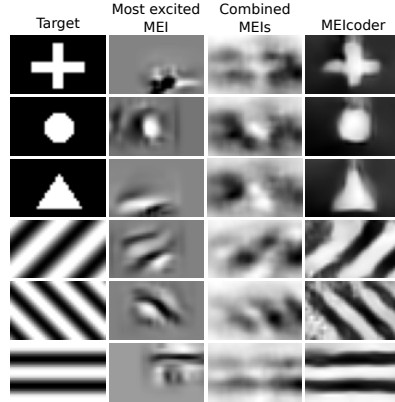

Figure 9: Reconstructing artificial patterns using MEIs (center) and MEIcoder (right). Shown are only the middle image regions for better visibility of MEIs.

**Reconstructing artificial patterns.** In Figure 9, we provide an additional demonstration of how MEIcoder successfully generalizes to out-of-distribution data using its strong MEI prior. Specifically, we create images with artificial patterns for which we predict neuronal responses from CNN encoder pretrained on the BRAINREADER dataset, and then decode the original images back using these encoded responses. As we can see, MEIcoder captures exact shapes well, even though it has never encountered such artificial patterns in its training data. This demonstrates its generalization and reliability, which might be crucial for certain applications. In addition, we illustrate the underlying intuition behind our method: approximating the original images by summing the MEIs of all neurons, weighted by their corresponding neuronal responses ("Combined MEIs"). Additionally, we also show the MEI corresponding to the neuron with the highest predicted response ("Most excited MEI"). The fact that even these simple linear MEI-based reconstructions already capture some of the basic characteristics of the original images further motivates the main building block of MEIcoder.

**Concept-based analysis.** Finally, we analyze the learned decoding process of MEIcoder, aiming to provide scientific insights into computations in V1. Specifically, we implement a concept-based analysis that combines (1) non-negative matrix factorization (NMF) to learn a dictionary of 32 feature bases ("visual concepts") from the feature maps at different layers of the MEIcoder's core module, with (2) a sensitivity analysis to observe how manipulating the response of a single neuron changes the activation of these concepts. Intuitively, if the increased neuron activation increases the activation of certain concepts in our decoder, this would mean that in the brain, these visual concepts are likely to be driving that particular neuron. We make the following observations from this analysis:

1. For the majority of neurons, the high-intensity (bright) areas of the most active concepts become smaller and more focused as we traverse through the decoder's layers. This suggests the model learns a hierarchy, starting with coarse features and progressively refining them into more detailed structures, with the final location remaining in most cases consistent with the neuron's original MEI. An example of this for two neurons can be seen in Figure 10.

2. Many of the neurons' top three feature bases include a concept that encodes the brightness of the image border (Figure 10). Together with finding (1), this suggests that many neurons encode the global lighting condition, and at the same time specialize in encoding smaller local structures at different places in the visual field (as supported by MEIs). This might hint at how, through lateral cortical processing, the cortex fills in information where it is missing.

3. We identify a few neurons for which incrementally increasing the response results in an incremental shift of a dark object in the reconstructed image (two examples in Figure 12). This highlights neurons whose responses drastically affect the decoder's learned reconstruction process. Interestingly, this emergent MEIcoder's property mirrors key findings about functional asymmetries in the visual cortex. Namely, a study by [49] found a significant over-representation of "black-dominant" (OFF) neurons in the corticocortical output layers 2/3 of macaque V1. As the authors of [49] pointed out, their results suggested that the human perceptual preference for black over white is generated or greatly amplified in V1.

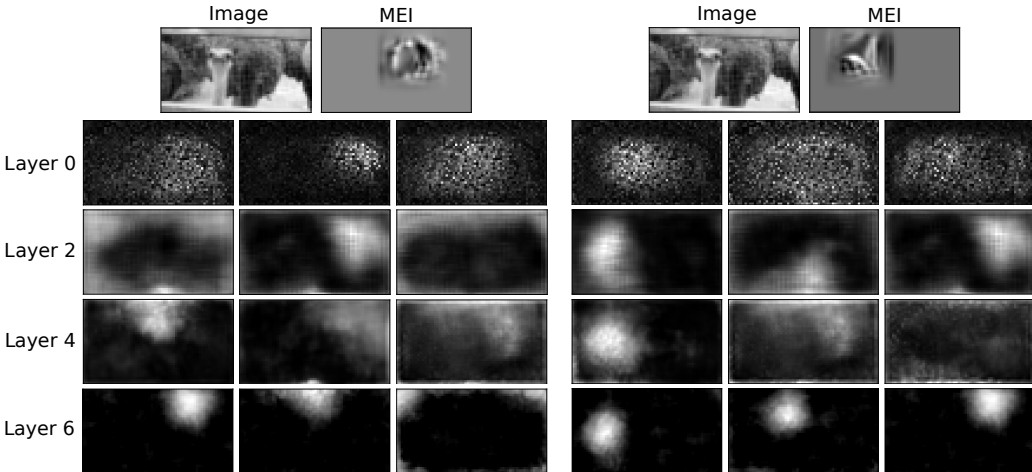

Figure 10: Input images, MEIs, and visual concepts with the highest activation gain for two example neurons from the BRAINREADER dataset. For each MEIcoder core layer and neuron, the top three concepts (NMF features) are shown, ordered by the increase in the corresponding feature coefficient between a forward pass with the neuron's response set to zero and one with it set to $10 \cdot p_{.99}$, where $p_{.99}$ denotes the neuron's 99th activation percentile in the dataset.

This analysis showcases MEIcoder's potential as a tool for scientific discovery, revealing how it combines individual neuron contributions into a coherent reconstruction. This demonstrates its utility for studying visual neural codes, and we believe further interpretability analysis represents an exciting avenue for future work. A more detailed discussion of the visual features reconstructed by MEIcoder and how our analysis compares with existing work can be found in subsection A.10.

## 5 Concluding remarks

**Limitations.** Motivated by applications in brain-machine interfaces and neuroprosthetics for individuals with acquired blindness, we focused primarily on data from V1, which encodes low-level features of visual stimuli. Although we showed state-of-the-art performance on three V1 datasets from two different species, which we would argue is still a remarkable feat, we did not test our method on higher-order areas such as V4. However, despite the increased complexity of receptive fields of neurons in these areas, MEIcoder can combine MEIs in a highly nonlinear fashion thanks to its core, and is able to learn additional coding properties of neurons in its learnable neuron embeddings. Another limitation of this study and room for additional investigation lies in the transfer learning capability of MEIcoder (i.e., pre-training on multi-subject data). Empirically, we found performance gains from transfer on the BRAINREADER dataset, but not on the SENSORIUM 2022 data.

**Conclusion.** We introduced MEIcoder, a novel decoding method that achieves state-of-the-art performance in reconstructing visual stimuli from neural population activity in V1. Leveraging MEIs, SSIM-based training objective, and adversarial training, MEIcoder outperforms all baselines on three difficult datasets, especially excelling in data- and neuron-scarce settings. Compared to GenAI techniques that gained popularity in decoding, MEIcoder better captures low-level features of images and does not suffer from high variance of reconstruction accuracy, making it more suitable for applications ranging from brain machine interfaces to uncovering brain information content. Our additional investigations also offer practical insights for neuroscience: We showed for the first time that MEIs are a powerful tool not only for understanding tuning properties of individual neurons, but also for reconstructing stimuli. Additionally, our scaling experiments demonstrate that it is feasible to achieve high-fidelity image reconstructions with as few as 1,000 to 2,500 neurons and limited single-subject training data. Finally, we showed the promise of accurate spiking models for developing decoding methods and presented an integrated benchmark pipeline that provides access to more than 160,000 samples to support future research in this area.

## Acknowledgments

JS was supported by the Bakala Foundation during his studies at EPFL. LB and JA were supported by the EU Horizon 2020 Maria SklodowskaCurie grant agreement No 861423. JA was supported by the ERDF-Project Brain dynamics, No. CZ.02.01.01\00\22_008\0004643 and by the grant no. 25-18031S of the Czech Science Foundation (GAČR).

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

# A  Technical Appendices and Supplementary Material

## A.1  Experimental details

We conducted the main experiments reported in Table 1 and Table 2 three times with different random seeds; the resulting standard deviations are reported in Table 3 and Table 4. Each of our experiments used one NVIDIA Tesla V100 GPU and required less than 32 GB of VRAM. The longest training run of our method (training with data from multiple subjects) took approximately four days, whereas the longest training run of the baselines took six days.

The code for our experiments and instructions on obtaining the data are available in our public repository at `https://github.com/Johnny1188/meicoder`.

Table 3: Quantitative results on the test sets from the BRAINREADER and SENSORIUM 2022 datasets. Best results are highlighted in red, and second-best in **bold**.

| Method | BRAINREADER | | | | SENSORIUM 2022 | | | |
|---|---|---|---|---|---|---|---|---|
| | SSIM | PixCorr | Alex(2) | Alex(5) | SSIM | PixCorr | Alex(2) | Alex(5) |
| InvEnc | .321 ±.001 | .611 ±.006 | .989 ±.002 | .896 ±.011 | .288 ±.005 | .453 ±.005 | .915 ±.010 | .720 ±.012 |
| EGG | .256 ±.006 | .495 ±.001 | .758 ±.015 | .659 ±.021 | .256 ±.009 | .365 ±.020 | .777 ±.007 | .755 ±.024 |
| MonkeySee | .232 ±.008 | .565 ±.007 | .967 ±.006 | .826 ±.024 | .185 ±.006 | .338 ±.009 | .564 ±.014 | .523 ±.004 |
| CAE | .256 ±.010 | .638 ±.003 | .930 ±.004 | .730 ±.007 | .287 ±.006 | **.539** ±.008 | .656 ±.005 | .549 ±.002 |
| MindEye2 | .277 ±.039 | .560 ±.027 | .946 ±.020 | .878 ±.027 | .210 ±.028 | .471 ±.036 | .877 ±.037 | .762 ±.048 |
| MindEye2 (FT) | .234 ±.045 | .516 ±.038 | .920 ±.043 | .836 ±.067 | .243 ±.006 | .499 ±.008 | .918 ±.013 | .799 ±.013 |
| MEIcoder | **.400** ±.022 | **.679** ±.010 | **.998** ±.002 | **.990** ±.006 | **.331** ±.004 | **.503** ±.013 | **.988** ±.004 | **.896** ±.006 |
| MEIcoder (FT) | **.424** ±.006 | **.706** ±.012 | **.999** ±.002 | **.977** ±.010 | **.318** ±.003 | .486 ±.011 | **.975** ±.004 | **.908** ±.003 |

## A.2  Data

Each data point consists of a z-scored image and associated neuronal responses (neural activity accumulated during ±500 ms time window after image stimulus onset). All images, except the hand-selected ones in the test sets of BRAINREADER and SENSORIUM 2022, were randomly sampled from ImageNet. The final sizes of the grayscale-mapped stimuli were: $36 \times 64$ px (BRAINREADER), $22 \times 36$ px (SENSORIUM 2022), and $20 \times 20$ px (SYNTHETIC CAT V1). Additionally, to address the substantial variability of firing rates between different neurons, which could be detrimental to training, we rescale individual neuronal responses by the inverse of the standard deviation estimated for each neuron. All z-scoring statistics are obtained from the training sets. For further details on data collection of the biological datasets, please refer to the original works ([11] and [48] for BRAINREADER and SENSORIUM 2022, respectively).

Table 4: Quantitative results on the test set of the SYNTHETIC CAT V1 dataset. Best results are highlighted in red, and second-best in **bold**.

| Method | SYNTHETIC CAT V1 | | | |
|---|---|---|---|---|
| | SSIM | PixCorr | Alex(2) | Alex(5) |
| InvEnc | **.771** ±.002 | **.833** ±.002 | **.986** ±.003 | **.978** ±.001 |
| EGG | .640 ±.022 | .667 ±.027 | .936 ±.007 | .906 ±.013 |
| MonkeySee | .607 ±.031 | .723 ±.019 | .982 ±.009 | .958 ±.021 |
| CAE | .637 ±.006 | **.793** ±.005 | .929 ±.004 | .775 ±.006 |
| MindEye2 | .559 ±.011 | .757 ±.016 | .977 ±.006 | .939 ±.014 |
| MEIcoder | **.774** ±.006 | .777 ±.009 | **.994** ±.000 | **.987** ±.002 |

**Synthetic Cat V1 dataset.** We generated the SYNTHETIC CAT V1 dataset using the biologically realistic spiking model from [3], which represents cortical layers 4 and 2/3, corresponding to a $5 \times 5$ mm patch of cat V1. When generating the samples, we first presented this encoding model with an image stimulus and then measured the evoked mean firing rates of 46,875 neurons in layer 2/3 over the following 560 ms time window. As for the other datasets, the image stimuli were sampled from ImageNet, converted to grayscale, downsampled, and then cropped to a size of $50 \times 50$ px. When we subsequently used the data to train and evaluate the decoding models, we only considered the central $20 \times 20$ px patch of the images. The reason is that the neurons in the encoding model have overlapping receptive fields that do not cover the whole visual field; therefore, their induced responses contain information only about the central patch of the presented images. To reduce the impact of noise on our evaluation, we created the test set by presenting the image stimuli 100 times and then averaging the corresponding neural responses to obtain the final neural activity.

Combining the three datasets into a single data corpus results in a benchmark pipeline consisting of:

1. BRAINREADER dataset: Data from V1 of 22 mice, where each single-subject dataset contains around 5,000 data points, and the average/min/max number of neurons is 8,116/6,721/9,395. Data for each mouse was split into training, validation, and test sets by the original authors [11]. The test sets contain 40 repeated trials for each stimulus.

2. SENSORIUM 2022 dataset: Data from V1 of 5 mice (using only the *training recordings* from the SENSORIUM 2022 competition), where each single-subject dataset contains around 5,000 data points, and the average/min/max number of neurons is 7,851/7,334/8,372. We split the dataset from each mouse into a training (4,500) and a validation (500) set. The test sets are provided separately by the original authors [48] and contain 10 repeated trials for each stimulus.

3. SYNTHETIC CAT V1 dataset: Synthetic data from the spiking model of cat V1 (single-subject) containing 50,250 data points of neuronal responses of 46,875 neurons. We split this data corpus into a training (45,000), validation (5,000), and test (250) sets. The test set contains 100 repeated trials for each stimulus.

### A.3 MEIcoder details

**Hyperparameters.** We provide hyperparameters of MEIcoder used for the final experiments in Table 5. Additional settings that we kept the same across all datasets include:

- Number of compressed neural map channels $d_c$ (readin): 480
- Number of CNN channels (core): 480, 256, 256, 128, 64, 1
- Kernel sizes (core): 7, 5, 5, 3, 3, 3
- Padding (core): 3, 2, 2, 1, 1, 1
- Stride (core): 1, 1, 1, 1, 1, 1
- Dropout probability (core): 0.35

Table 5: Hyperparameter search space for MEIcoder, with final selected values underlined.

| Dataset | Learning rate | Weight decay | Dimension of neuron embeddings |
|---------|--------------|--------------|-------------------------------|
| BRAINREADER | {1e-3, 1e-4, 3e-5} | {3e-1, 8e-2, 3e-3} | {16, 32, 64} |
| SENSORIUM 2022 | {1e-3, 1e-4, 3e-5} | {3e-1, 8e-2, 3e-3} | {16, 32, 64} |
| SYNTHETIC CAT V1 | {1e-3, 1e-4, 3e-5} | {3e-1, 8e-2, 3e-3} | {16, 32, 64} |

**Discriminator.** The discriminator used for the auxiliary adversarial objective is implemented as a CNN with five layers of convolution, batch normalization, ReLU activation function, and dropout ($p = 0.3$). The output of the last layer is flattened into a one-dimensional vector and transformed by a linear layer followed by the sigmoid activation function. The result is a predicted probability that the given discriminator's input is a reference image from the dataset (i.e., not a reconstruction from the decoder).

We train the discriminator simultaneously with the decoding model using the AdamW optimizer [26] with the same learning rate and weight decay as the decoding model. The training objective is as follows: Let $\lambda_{\text{GT}} \in \mathbb{R}_+$ be the loss weighting factor, $\epsilon_{\text{GT}} \in [0, \xi_{\text{GT}} \in \mathbb{R}_+]$, $\epsilon_{\text{R}} \in [0, \xi_{\text{R}} \in \mathbb{R}_+]$ be the target noising components, and $\mathbf{y}$ and $\hat{\mathbf{y}}$ denote the reference (ground-truth) and reconstructed image, respectively. Then, our implementation of the discriminator loss $\mathcal{L}_{\text{D}}$ is the following:

$$\mathcal{L}_{\text{D}}\big(\mathbf{y}, \hat{\mathbf{y}}\big) = \lambda_{\text{GT}} \cdot \big(\mathbf{D}_\phi(\mathbf{y}) - 1 - \epsilon_{\text{GT}}\big)^2 + (1 - \lambda_{\text{GT}}) \cdot \big(\mathbf{D}_\phi(\hat{\mathbf{y}}) - \epsilon_{\text{R}}\big)^2. \tag{3}$$

The target noising component $\epsilon_{\text{GT}}$ for the reference image part is sampled from a uniform distribution between 0 and $\xi_{\text{GT}}$, while the noising component $\epsilon_{\text{R}}$ comes from a uniform distribution between 0 and $\xi_{\text{R}}$. We found that by introducing noise into the discriminator training, we were able to better balance the decoder and the discriminator, and thereby stabilize the training. We note that this is reminiscent of *one-sided label smoothing* as introduced in [34], where the discriminator's positive targets are smoothed from 1 to 0.9, making its task harder. The specific hyperparameters we used for the final experiments are given below:

- Number of channels: 256, 256, 128, 64, 64
- Kernel sizes: 7, 5, 3, 3, 3
- Padding: 2, 1, 1, 1, 1
- Stride: 2, 2, 1, 1, 1
- $\xi_{\text{GT}} = \xi_{\text{R}} = 0.05$
- $\lambda_{\text{GT}} = 0.5$

Overall, the selection of architecture and hyperparameters makes MEIcoder more parameter-efficient. Namely, the CNN architecture achieves efficiency by parameter-sharing and by allowing only local connections. Furthermore, MEIcoder reuses the core module (backbone) across datasets from different subjects (e.g., different mice) and only trains new readin modules. By sharing the core module, MEIcoder reduces the number of parameters threefold.

### A.4 Most exciting inputs

**Motivation.** The two main reasons why we decided to use a computational prior in the form of MEIs are the following. First, strong priors help guide learning and prevent overfitting in data-limited regimes like our neural recording dataset [5, 7]. Second, many neurons exhibit sparse firing, which can make it difficult for the decoder to learn the stimulus-response mapping for a neuron that was active for only a handful of images in the training set. To combat this, the MEI provides a powerful "head-start" by giving the decoder a dense, explicit template of each neuron's preferred stimulus, even for rarely firing neurons. This is visually demonstrated in Figure 9, where simply weighting MEIs by neural responses already forms a coarse but recognizable reconstruction, highlighting the power of this prior.

**Generation.** We follow previous work [4, 32, 45] to generate MEIs of all neurons in the given single-subject dataset. More specifically, we train a CNN-based encoding model on the given dataset, randomly initialize an MEI image with zero mean and standard deviation of 0.15, and then iteratively optimize its pixel values using gradient ascent to maximize the encoder's prediction for the target neuron. After each optimization step, we normalize the image back to zero mean and a standard deviation of 0.15 and clip pixel values that have an absolute value larger than one. Since there are more than 7,000 neurons in each dataset, we accelerate this MEI generation procedure by initializing and then optimizing MEIs of multiple neurons in parallel (batching inputs to the encoder). Note that this whole optimization procedure needs to be done only once for each dataset.

While the entire MEI generation, including encoder training, took approximately 70 minutes (15 minutes training, 55 minutes generation) in our experiments, the decoder training required 11 hours on one NVIDIA Tesla V100 GPU. This shows that the computational overhead of MEI generation is small compared to the decoder training itself.

For all use cases of CNN-based encoding model (MEIs, InvEnc, and EGG), we use the *Gaussian readout* architecture introduced by [27]. More specifically, we use the implementation and hyperparameters provided by [48].

## A.5 Sensitivity to quality of most exciting inputs

To understand the sensitivity of MEIcoder's performance to the quality of generated MEIs, we re-ran training and evaluation on the BRAINREADER dataset with MEIs with varying levels of Gaussian noise. As shown in Table 6, the performance degrades relatively slowly. In fact, even with highly noisy MEIs (std=1 and std=3 for MEIs with initial pixel values between -1 and 1), the performance remains on par or better than that of the baselines (Table 1).

Table 6: Quantitative results on the test set of the BRAINREADER dataset. Standard deviation (std) corresponds to the Gaussian noise added to MEIcoder's MEIs.

| Method | BRAINREADER | | | |
|---|---|---|---|---|
| | SSIM | PixCorr | Alex(2) | Alex(5) |
| MEIcoder (no noise) | .400 | .679 | .998 | .990 |
| MEIcoder (std=0.2) | .402 | .675 | .997 | .982 |
| MEIcoder (std=0.5) | .390 | .670 | .990 | .938 |
| MEIcoder (std=1) | .332 | .625 | .990 | .937 |
| MEIcoder (std=3) | .287 | .568 | .984 | .943 |

## A.6 Comparison to gradient-based linear receptive fields of neurons

MEIs can be seen as a nonlinear counterpart to traditional linear receptive fields of neurons. To compare these neural characterizations for decoding, we replaced MEIs in MEIcoder with gradient-based linear receptive fields (LRFs) obtained from regularized regression trained on neural responses from the BRAINREADER dataset. As shown in Table 7, this leads to degraded performance, but not as severe as with a complete removal of neural characterization as done in the initial ablation study in subsection 4.5. This demonstrates the importance of biologically informed computational prior, MEIs in particular, for the decoder's performance.

Table 7: Quantitative results on the test set of the BRAINREADER dataset with different neural characterizations in MEIcoder's readin.

| Method | BRAINREADER | | | |
|---|---|---|---|---|
| | SSIM | PixCorr | Alex(2) | Alex(5) |
| MEIcoder (MEIs) | .400 | .679 | .998 | .990 |
| MEIcoder (LRFs) | .364 (-9%) | .663 (-2.4%) | .998 (-0%) | .949 (-4.1%) |

## A.7 MEIcoder for highly non-linear neurons

To demonstrate that the state-of-the-art performance of MEIcoder does not severely degrade when trying to decode from highly non-linear neurons, such as those found in higher visual areas of the brain, we conducted the following comparative experiment.

First, for each neuron in the BRAINREADER dataset, we calculated the so-called non-linearity index (NLI), which estimates how non-linear individual neurons are [2]. It is calculated as the ratio between the prediction power of a linear encoding model fitted to the data and the prediction power of a state-of-the-art non-linear encoding model fitted to the data. Second, we trained and evaluated MEIcoder only on subsets of the most linear and most non-linear neurons. As can be seen in Table 8, MEIcoder's ability to decode images from the more non-linear neurons is very similar to its ability to decode from more linear neurons, suggesting that MEIcoder can handle non-linearity of the code very well, at least within the context of V1.

Table 8: Quantitative results on the test set of the BRAINREADER dataset when MEIcoder is trained and evaluated only with subsets of neurons.

| Selection of neurons | BRAINREADER | | | |
|---|---|---|---|---|
| | SSIM | PixCorr | Alex(2) | Alex(5) |
| 3,000 most non-linear neurons | .321 | .630 | .999 | .958 |
| 3,000 least non-linear neurons | .324 | .653 | .996 | .939 |
| 3,000 least non-linear neurons (nonzero NLI) | .345 | .663 | .999 | .979 |

Table 9: Quantitative results on the test sets from the BRAINREADER and SENSORIUM 2022 datasets (higher-level metrics). Best results are highlighted in red, and second-best in **bold**.

| Method | BRAINREADER | | | | SENSORIUM 2022 | | | |
|---|---|---|---|---|---|---|---|---|
| | Incep ↑ | CLIP ↑ | Eff ↓ | SwAV ↓ | Incep ↑ | CLIP ↑ | Eff ↓ | SwAV ↓ |
| InvEnc | .627 | .636 | .606 | .448 | .600 | .564 | .492 | .243 |
| EGG | .749 | .610 | .652 | .497 | .673 | .579 | .449 | .232 |
| MonkeySee | .655 | .547 | .653 | .486 | .614 | .513 | .551 | .301 |
| CAE | .592 | .541 | .628 | .612 | .542 | .535 | .684 | .737 |
| MindEye2 | .772 | .636 | **.571** | **.415** | .643 | .590 | **.440** | .235 |
| MindEye2 (FT) | .725 | .652 | .584 | .465 | .650 | **.595** | .443 | **.219** |
| MEIcoder | **.799** | **.679** | .586 | .489 | **.727** | .590 | **.440** | .276 |
| MEIcoder (FT) | **.817** | **.702** | **.520** | **.408** | **.746** | **.620** | **.419** | **.216** |

## A.8 Evaluation on higher-level metrics

Using the implementation from Mind-Eye2 [36], we evaluated all methods on higher-level (semantic) metrics. Specifically, we measured the two-way identification accuracy with InceptionV3 ("Incep") and CLIP (ViT-L/14) embeddings, as well as the average correlation distance in the embedding space of EfficientNet-B1 ("Eff") and SwAV-ResNet50 ("SwAV").

As can be seen in Table 9 and Table 10, ME-Icoder remains highly competitive even on these semantic metrics (Incep and CLIP: the higher the better, Eff and SwAV: the lower the better). On both the BRAINREADER and SENSORIUM 2022 datasets, MEIcoder achieves the best performance, while on the

Table 10: Quantitative results on the test set of the SYNTHETIC CAT V1 dataset as measured on higher-level metrics. We highlight the best score in red and the second-best score in **bold**.

| Method | SYNTHETIC CAT V1 | | | |
|---|---|---|---|---|
| | Incep ↑ | CLIP ↑ | Eff ↓ | SwAV ↓ |
| InvEnc | **.884** | **.827** | .326 | .196 |
| EGG | .766 | .675 | .283 | .214 |
| MonkeySee | .796 | .760 | **.272** | **.193** |
| CAE | .657 | .622 | .442 | .573 |
| MindEye2 | .798 | **.780** | **.274** | .208 |
| MEIcoder | **.898** | .761 | .277 | **.181** |

SYNTHETIC CAT V1 data, it outperforms the other baselines on two out of four metrics, remaining on par on the other two performance measures. We can also see that the MEIcoder fine-tuned ("FT") from multi-subject to single-subject data performs the best on the biological datasets. This further demonstrates MEIcoder's ability to generalize across data from different subjects.

## A.9 Reconstructing higher-resolution images

Here, we demonstrate the scalability of MEIcoder to higher-resolution visual stimuli. Specifically, using the MEIs generated for the original resolution, we train and test MEIcoder on the BRAINREADER dataset with a two-times higher image resolution of $72 \times 128$ pixels.

As shown in Table 11, MEIcoder's performance is maintained at this image resolution. In fact, although the task of reconstructing larger images is inherently more difficult, the higher-resolution ME-Icoder still outperforms the best alternative baseline from the low-resolution setting (Table 1). This demonstrates that MEIcoder is not limited to low-resolution stimuli and can effectively scale to more detailed visual inputs.

Table 11: Quantitative results of MEIcoder on the test set of the BRAINREADER dataset.

| Image resolution | BRAINREADER | | | |
|---|---|---|---|---|
| | SSIM | PixCorr | Alex(2) | Alex(5) |
| $36 \times 64$ px | .400 | .679 | .998 | .990 |
| $72 \times 128$ px | .340 | .646 | .997 | .980 |

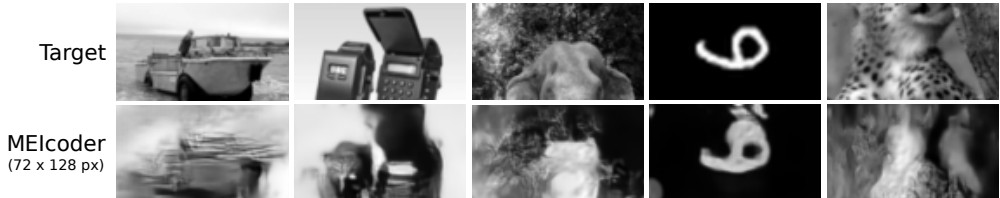

Figure 11: Reconstructions of images from the BRAINREADER dataset with MEIcoder trained and evaluated on $72 \times 128$ px images.

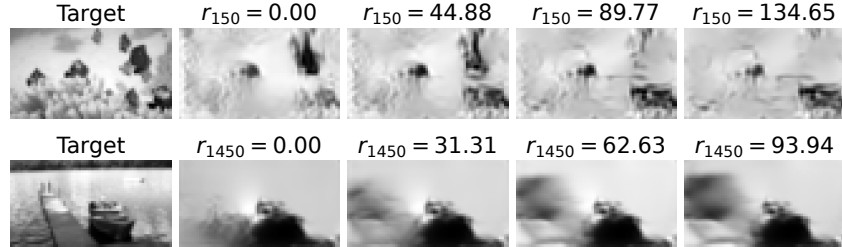

Figure 12: Reconstructions of images from the BRAINREADER dataset as we vary the response level of neurons 150 (top) and 1450 (bottom).

## A.10 MEIcoder interpretability

Our analysis in subsection 4.5 shares methodological tools with methods like CRAFT [14], particularly the use of NMF and sensitivity analysis. The main difference between CRAFT and our analysis is that we employ sensitivity analysis to derive the importance of inputs (neuronal responses) on intermediate features (coefficients for combining NMF concepts). CRAFT, on the other hand, uses sensitivity analysis to assess the importance of intermediate features (coefficients for combining NMF concepts) on the final output of the network. To measure the importance of inputs on the intermediate features, CRAFT employs implicit differentiation and gradient-based attribution maps.

Other techniques, such as ACE [16], ICE [50], and sparse autoencoders (SAEs) [15, 20] have explored similar ideas but on different tasks and with different methodologies. ACE, for example, considered discovering visual features by segmenting images from the same semantically meaningful class and clustering these segments in an embedding space of a CNN. Then, akin to our sensitivity analysis, ACE perturbed the hidden states to evaluate the importance of individual segment clusters on the final prediction of the CNN classifier.

**Reconstructed visual features.** Analyzing which visual features are reconstructed with high fidelity can tell us which of them are well-encoded in the neural population of V1. Here, we find two key patterns.

First, MEIcoder's reconstruction performance directly reflects the known tuning properties of V1 neurons: it consistently reconstructs distinct, high-contrast features like corners and edges with high fidelity, while struggling with low-frequency information such as gradual changes in shading (see, for example, figures 2 and 15). This is a reflection of the V1 code, which is dominated by edge-detecting neurons that provide a sparse signal for uniform surfaces.

Second, we see a more global failure mode where images containing dense, high-frequency textures across a relatively small area of the image lead to a globally degraded reconstruction, an effect especially pronounced in the SENSORIUM 2022 dataset (Figures 2 and 14). This suggests that while V1 robustly encodes local details, the decoder can be "overwhelmed" when trying to synthesize a coherent global percept from a highly complex population signal, hinting at why the brain requires hierarchical processing.

## A.11 Broader impacts

Our work advances neural decoding by enabling high-fidelity image reconstruction from limited V1 neural activity, offering insights into visual processing and potential applications in brain-machine

interfaces. With its data- and neuron-efficiency, MEIcoder lowers the requirements on invasive brain recordings, which we believe is of great importance for practical neuroengineering applications.

However, translating our findings to human applications requires caution, as models trained on constrained and multi-subject datasets may inherit biases, potentially limiting generalization across diverse populations. Therefore, future work should incorporate broader, more inclusive datasets and rigorous clinical validation to ensure high performance, balancing scientific progress with ethical responsibility.

## A.12 Additional reconstructed images

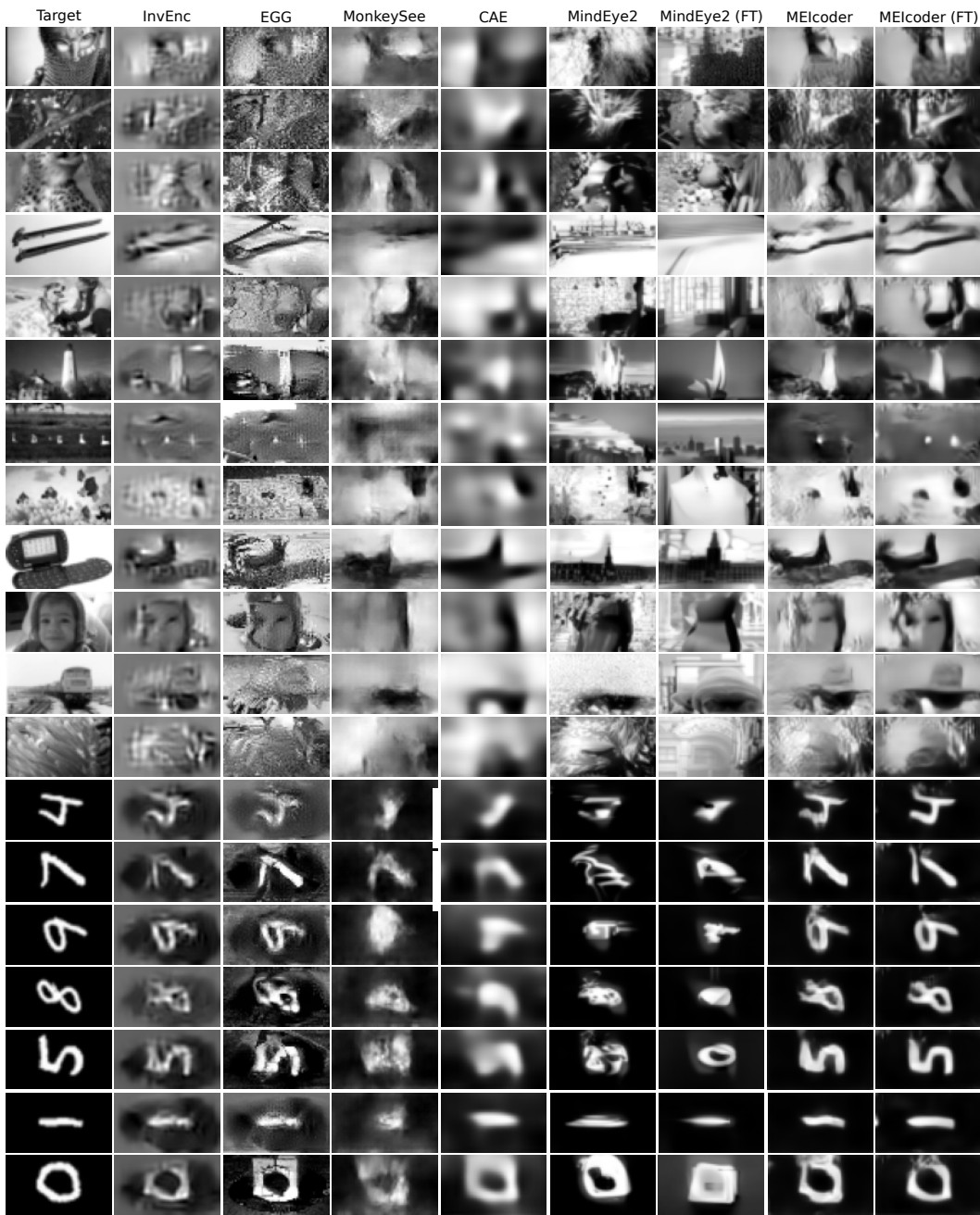

Figure 13: Additional reconstructed images from the BRAINREADER dataset.

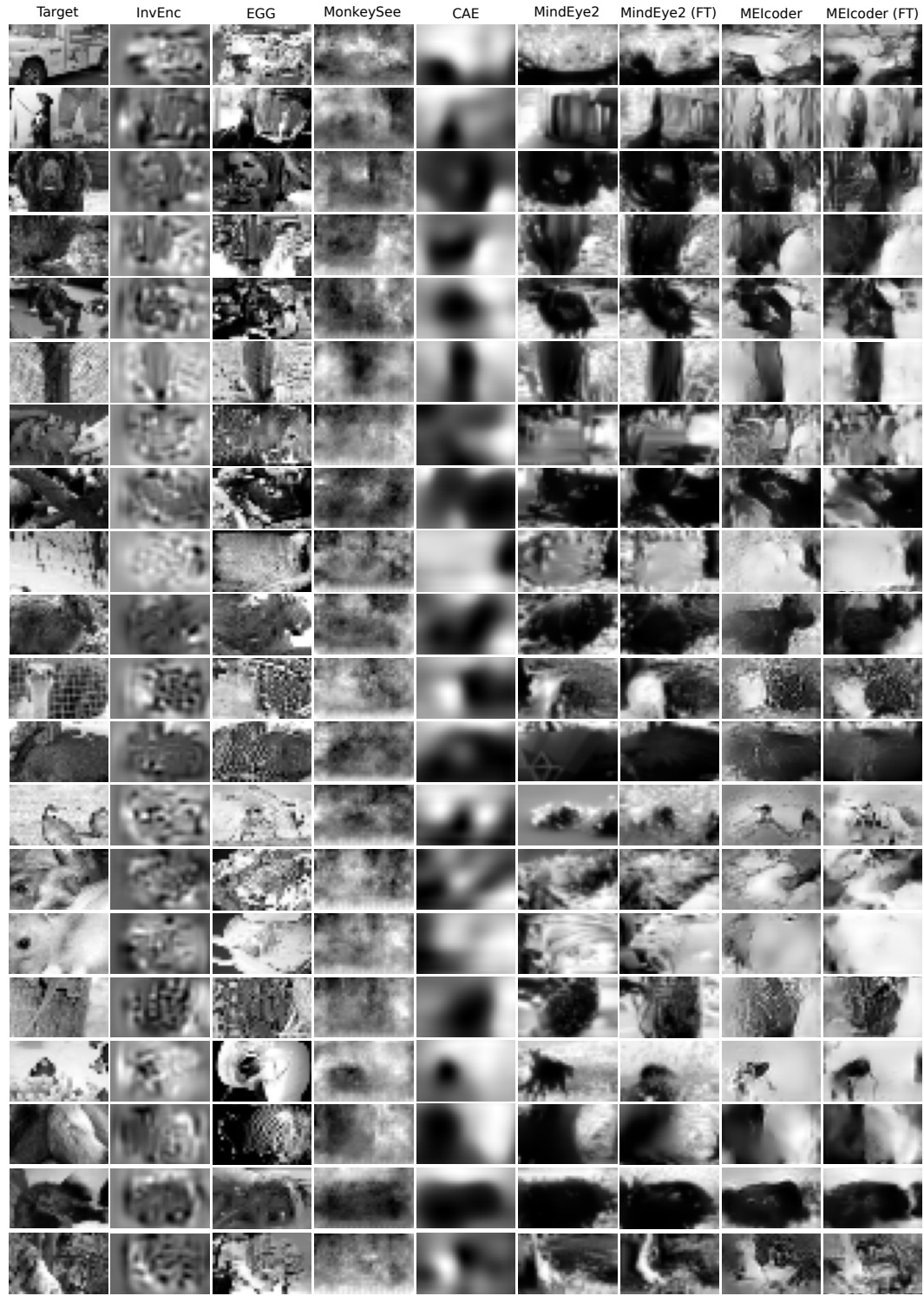

Figure 14: Additional reconstructed images from the SENSORIUM 2022 dataset.

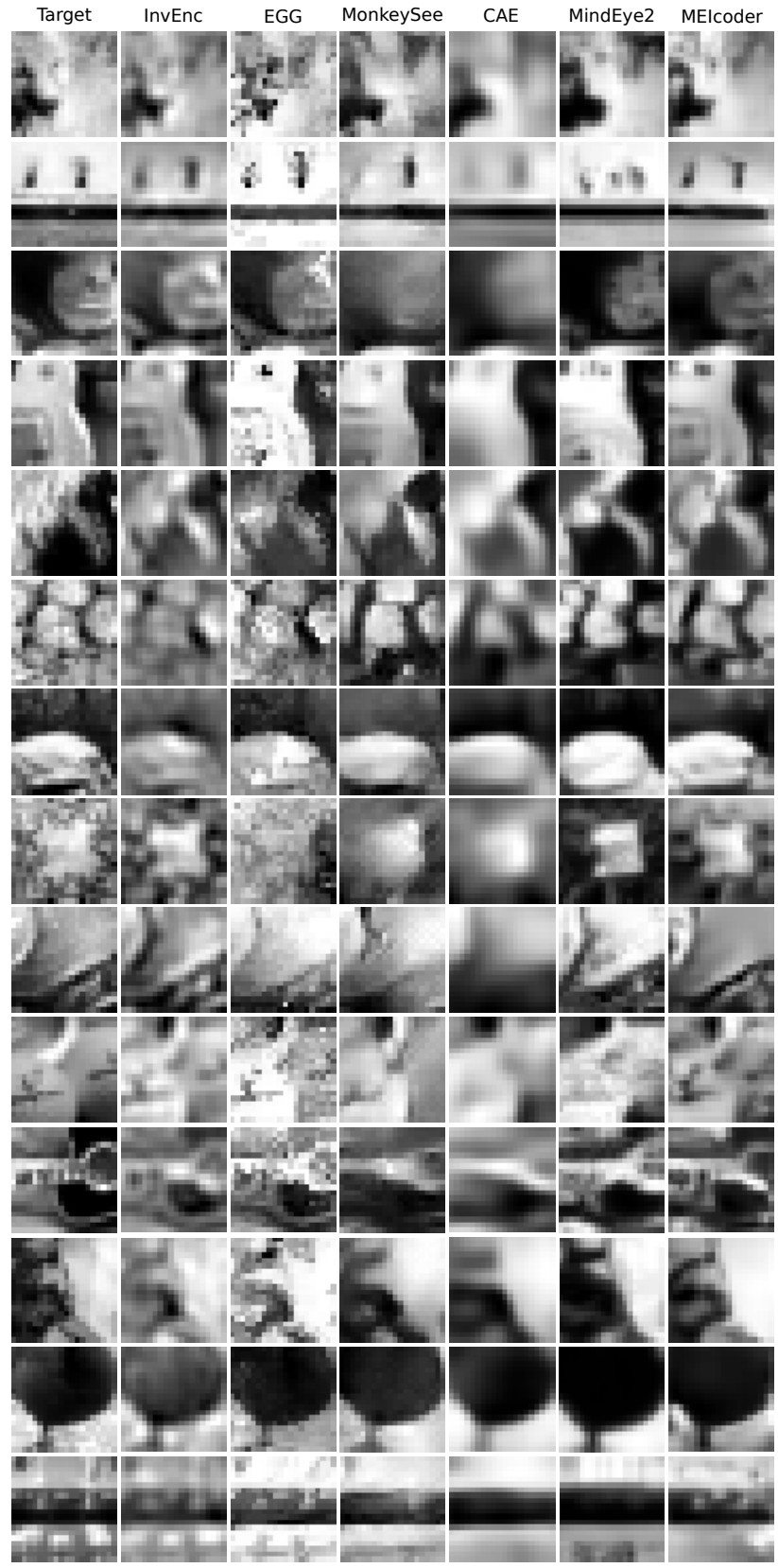

Figure 15: Additional reconstructed images from the SYNTHETIC CAT V1 dataset.

