# OpenReview forum: "MEIcoder: Decoding Visual Stimuli from Neural Activity by Leveraging Most Exciting Inputs"
_NeurIPS.cc/2025/Conference — NeurIPS 2025 poster_

### Official Review · Reviewer_iULc · 2025-06-11

**Clarity:** 4
**Significance:** 4
**Originality:** 4
**Rating:** 5
**Confidence:** 4

**Summary:**

This paper uses a mixture of expert knowledge, and different techniques from machine learning to study the inverse problem of taking neuronal activity and converting it to observed images. This problem is a important and complex problem for the development of prosthesis for those with vision loss or vision impairment. This work takes a non-intuitive leap in the design and approach in this problem area by focusing on most exciting inputs, which appear to play an important role in how images are encoded by the eye to neuronal signals. The ability to replicate this behavior through deep learning processes efficiently is an important computational leap. The combination of hybrid techniques used to do this replication indicate that the vision encoding problem is not simple or straightforward and is actually quite bottlenecked when it comes to energy efficiency. The presentation of the work is acceptable. The techniques are rather novel for neuroscience, and do give some evidence for related problems in computer science.

**Questions:**

Due to limited time in my reviewing schedule, I don't have any further questions.

**Ethical Concerns:**

["NO or VERY MINOR ethics concerns only"]

**Final Justification:**

I maintain my score.

**Quality:**

4

**Strengths And Weaknesses:**

Strengths:

Very good empirical study

The inverse problem is less looked at, but actually more important for making BCIs as the eventual purpose of BCIs is to *stimulate* neurons as if they were being stimulated by human appendages.

The usage of a computational prior is good, but needs to be better justified. Why is a computational prior needed for this exact problem?

The development of a dataset for further work is a strong contribution.

Section 4.5 forms a good section for discussion that is more than typical for AI works.


Weaknesses:

After having carefully read the entire appendix, there are no ethical concerns.

The necessity or importance of using SSIMs need to be better explained.

The design choice of manifold and parameter efficiency needs to be better explained.

The related work section is acceptable, but does not unify the disparate works into a consistent story or narrative.

Section 3 is a bit dry to read.

---

> ### Author Rebuttal · Authors · 2025-07-31
>
> We thank the reviewer for their supportive review and comments. We are particularly grateful that the reviewer recognized the importance of our work for the development of brain-machine interfaces, the strength of our empirical study, and the value of our new benchmark dataset as a “strong contribution” to the community. The reviewer’s feedback primarily points to valuable opportunities to improve the clarity and justification of our methodological choices. We agree that these explanations will strengthen the paper, and in our response below, we provide these detailed justifications for the use of computational priors, our SSIM-based loss, and the parameter efficiency of our architecture, which we are confident will fully address the reviewer’s points.
>
> ---
>
> > **Strength 3:** *Usage of MEIs as a computational prior is good, but needs to be better justified. Why is a computational prior needed for this exact problem?*
>
> We thank the reviewer for pointing out the soundness of MEIs as a computational prior and their follow-up question, which gets to the heart of our motivation. A computational prior is particularly crucial in this setting for two main reasons. First, neural recording datasets are often data-limited, and deep learning models benefit from strong priors to prevent overfitting and guide learning [1,2]. Second, and more specific to this problem, many neurons exhibit sparse firing. A decoder may struggle to learn the stimulus-response mapping for a neuron that was active for only a handful of images in the training set. The MEI provides a powerful “head-start” by giving the decoder a dense, explicit template of each neuron’s preferred stimulus, even for rarely firing neurons. This is visually demonstrated in Figure 9, where simply weighting MEIs by neural responses already forms a coarse but recognizable reconstruction, highlighting the power of this prior. We will include this justification in the next revision of the paper.
>
> *[1] Christopher Bishop (2006). Pattern Recognition and Machine Learning. Springer.*
>
> *[2] Peter W. Battaglia, et al. (2018). Relational inductive biases, deep learning, and graph networks. arXiv preprint arXiv:1806.01261.*
>
> ---
>
> > **Weakness 1:** *Ethical concerns.*
>
> We thank the reviewer for their careful assessment. We have included a “Broader Impacts” section (A.6) in the original submission that discusses potential ethical considerations and societal impacts of this research line. If the reviewer considers this to be insufficient, please let us know, and we would be happy to expand on ethical concerns further.
>
> ---
>
> > **Weakness 2:** *Importance of SSIM.*
>
> We thank the reviewer for this comment. Our choice is motivated by our comparison with MSE (Section 4.5, ablation study) and SSIM’s proven ability to capture structural similarity in a way that aligns better with human perception than pixel-wise losses [3,4]. Since our primary goal is geometric fidelity for applications like visual prosthetics, a loss function that prioritizes perceptually coherent structures over exact pixel intensities is more appropriate. We will ensure this reasoning is stated more clearly in Section 3.1.
>
> *[3] Zhoung Wang, et al. (2004). Image quality assessment: from error visibility to structural similarity. IEEE Transactions on Image Processing, 13(4), 600-612.*
>
> *[4] Hamid Sheikh, et al. (2006). A Statistical Evaluation of Recent Full Reference Image Quality Assessment Algorithms. IEEE Transactions on Image Processing (Volume: 15, Issue: 11).*
>
> ---
>
> > **Weakness 3:** *The design choice of manifold and parameter efficiency needs to be better explained.*
>
> We thank the reviewer and apologize for the lack of clarity. Regarding parameter efficiency, we will clarify that it refers to the following two properties of MEIcoder:
> 1. MEIcoder employs a CNN architecture, which achieves efficiency by sharing parameters across different spatial locations within a feature map, and by using only local connections within its layers, as opposed to fully-connected networks.
> 2. MEIcoder reuses the core module across datasets from different subjects (e.g., 8 different mice from Brainreader) and trains only new subject-specific readin modules. For example, on the Brainreader dataset, the core module has around 18.5M parameters, and a single readin module contains around 4.5M parameters. If we trained both the core and the readin for each of the 8 mice in the Brainreader dataset separately, we would need around 184M parameters. By reusing the same core module across subjects, MEIcoder is able to lower this from 184M to just 54.5M.
>
> We are uncertain what the reviewer meant by “manifold”; if this refers to the latent space of our model, we will add more details on its structure in our new interpretability section. We would be grateful for any clarification the reviewer could provide on this point.
>
> ---
>
> > **Weakness 4:** *Related work section.*
>
> We appreciate this feedback. We will revise the related work section to not only list prior studies but also to synthesize them into a more cohesive narrative, highlighting the key challenges in the field (e.g., data limitation, geometric vs. semantic fidelity) and clearly positioning our work within this context.
>
> ---
>
> > **Weakness 5:** *Section 3 is a bit dry to read.*
>
> We thank the reviewer for this candid feedback. We will revise Section 3 to improve its readability, for instance, by adding more high-level motivation for our design choices and ensuring a clearer flow between the technical descriptions.

---

### Official Review · Reviewer_KLSf · 2025-06-29

**Clarity:** 3
**Significance:** 2
**Originality:** 2
**Rating:** 5
**Confidence:** 3

**Summary:**

This paper introduces MEIcoder, a method designed to reconstruct visual stimuli from neural population activity by leveraging Most Exciting Inputs. MEIcoder combines neuron-specific MEIs, structural similarity (SSIM)-based loss, and adversarial training to achieve state-of-the-art decoding performance across datasets from primary visual cortex (V1) recordings. Experiments on three datasets  demonstrate MEIcoder's superiority, particularly in data-scarce and neuron-constrained scenarios.

**Questions:**

Q1. Have you examined which specific visual features are consistently reconstructed with high fidelity versus those that are reconstructed poorly? Such analysis could provide valuable insights into V1 encoding mechanisms.

Q2. Have you considered exploring reconstructions through a concept-based approach, such as applying sparse dictionary learning on early layers of the model, to interpret and explicitly identify the visual features driving neuronal responses?

**Ethical Concerns:**

["NO or VERY MINOR ethics concerns only"]

**Final Justification:**

convincing rebuttal and adressing M1 and M2.

**Limitations:**

yes

**Paper Formatting Concerns:**

no formatting concerns

**Quality:**

2

**Strengths And Weaknesses:**

I appreciate this paper's attempt to bridge comp. neuro insights with practical decoding applications through MEI as biological priors. The work demonstrates clear technical merit and addresses a timely challenge in neural decoding.

Here are, to me, the main pros:

- Novel biological prior: The use of MEIs as structured priors is new and well-motivated, providing a principled way to inject neuroscience knowledge into the decoder
- Data efficiency: The method works remarkably well with limited data (<1,000 training samples) and few neurons (1,000-2,500), which has important practical implications
- Comprehensive evaluation: The paper includes thorough ablations, scaling experiments, and a valuable benchmark aggregating 160,000+ samples across datasets
- Strong empirical results: Clear improvements over baselines across multiple metrics and datasets, with particularly notable gains in data-scarce settings

However, I found several areas that could strengthen the contribution, i'll group them into Major (M) and minor (m):
Major Issues (M):

* M1. Limited scientific insights: While the technical achievement is solid, the paper would benefit from exploring what MEIcoder reveals about visual processing. Currently, it demonstrates capability without leveraging this to advance neuroscience understanding. A section analyzing reconstruction or to understand specific pattern or even better to motivate a new hypothesis (and test it) could provide valuable insights into V1 encoding principles.

* M2. Limited impact and generalization concerns: The evaluation is limited to V1 and relatively simple stimuli (small grayscale images). The work would be strengthened by discussing pathways to higher visual areas or more complex stimuli, even if not immediately feasible.

Now for the minor Issues (m):

* m1. MEI generation details: The circular dependency of needing a pre-trained encoder for MEI generation could be better addressed. How sensitive is performance to MEI quality?
* m2. Baseline comparisons: Some baselines perform surprisingly well (e.g., InvEnc on BRAINREADER). The paper would benefit from deeper analysis of why MEIcoder's advantages are more pronounced in certain settings.
* m3. Computational costs: No discussion of the computational overhead for MEI generation and the parameter-efficient architecture claims need more support.
* m4. Loss function justification: While SSIM performs well empirically, the paper could strengthen the theoretical motivation for this choice over alternatives (e.g., Feature visualization in XAI usually preconditioning in Fourier domain...)



**Overall**

This work makes a valuable contribution to neural decoding by incorporating biological priors through MEIs. The technical execution is solid, and the creation of a unified benchmark is commendable. The performance in data-scarce regimes addresses a real challenge in neuroscience applications.

However, the impact is currently limited by its focus on engineering improvements rather than scientific discovery. The paper has excellent potential to bridge this gap : the authors have built a tool that could reveal new insights about visual processing. By adding analysis of what reconstruction patterns tell us about V1 computation, or how biological and synthetic data compare, this could become a significant contribution to both ML and neuroscience.

I encourage the authors to view MEIcoder not just as an improved decoder, but as a lens for understanding visual processing. With modest additions exploring the neuroscience implications of their results, this could be an excellent paper that advances both fields.

---

> ### Author Rebuttal · Authors · 2025-07-31
>
> We thank the reviewer for their constructive and thoughtful feedback. We sincerely appreciate the reviewer for recognizing the novelty of our work in using MEIs as a principled biological prior, the practical importance of our method’s data efficiency in challenging data-scarce scenarios, and the rigor of our comprehensive evaluation. We also value the reviewer’s perspective that the creation of a unified benchmark with over 160,000 samples is commendable in itself. We address the reviewer’s comments below.
>
> ---
>
> > **M1, Question 2:** *Limited insights for neuroscience and concept-based analysis of reconstructions.*
>
> We agree that one of the key potential contributions of any neural decoder is the scientific insight it can offer. Our primary focus in this study was to develop a new method that would push the state-of-the-art performance forward and to assemble a new benchmark dataset so that future work in this field can proceed more rigorously. Nevertheless, following the reviewer's excellent suggestion in Question 2, we have implemented a concept-based analysis that combines **(1) Non-negative Matrix Factorization** (NMF) to learn a dictionary of 32 “visual concepts” from the feature maps at different layers of our decoder, with **(2) a sensitivity analysis** to observe how manipulating the response of a single neuron activates these specific concepts. Our key findings include:
> 1. For the majority of neurons, the high-intensity (bright) areas of the most active feature bases (concepts) become smaller and more focused as we traverse through the decoder’s layers. This suggests the model learns a hierarchy, starting with coarse features and progressively refining them into more detailed structures, with the final location remaining consistent with the neuron’s original MEI.
> 2. Many of the neurons’ top three feature bases include a concept that encodes the brightness of the image border. Together with finding (1), this suggests that many neurons encode the global lighting condition, and at the same time specialize in encoding smaller local structures and edges at different places in the visual field (as supported by MEIs). This might hint at how, through lateral cortical processing, the cortex fills in information where it is missing.
> 3. We identified a few neurons for which incrementally increasing the response results in an incremental shift of a dark object in the reconstructed image. This highlights neurons whose responses drastically affect the decoder’s learned reconstruction process. Interestingly, this emergent MEIcoder’s property mirrors key findings about functional asymmetries in the visual cortex. Namely, a study by [1] found a significant over-representation of “black-dominant” (OFF) neurons in the corticocortical output layers 2/3 of macaque V1. As the authors of [1] pointed out, their results suggested that the human perceptual preference for black over white is generated or greatly amplified in V1.
>
> This new analysis directly addresses the major concern (M1) about scientific insight and leverages MEIcoder as a “lens for understanding visual processing”, framing it as a promising avenue for future investigation.
>
> *[1] Chun-I Yeh, et al. (2009). “Black” responses dominate macaque primary visual cortex v1. The Journal of neuroscience: the official journal of the Society for Neuroscience, 29(38), 11753–11760.*
>
> ---
>
> > **M2:** *Limited impact and generalization concerns.*
>
> With regard to **higher-resolution stimuli**, we re-ran MEIcoder on the Brainreader dataset at the resolution of 72x128 pixels. As can be seen in the table below, our method’s strong performance holds in comparison to the lower-resolution version, demonstrating its applicability to more complex visual inputs. While we could not re-run all the baselines due to the limited time of the rebuttal period, we would like to point out that although the task of reconstructing larger images is inherently more difficult, the higher-resolution MEIcoder still achieves better performance than the best alternative baseline from the low-resolution setting (Table 1, original submission), strongly indicating that it will keep its competitive edge over the baselines also for the high-resolution condition.
>
> ||SSIM|PixCorr|Alex(2)|Alex(5)|
> |-|-|-|-|-|
> |**MEIcoder (36x64px)**|.400|.679|.998|.990|
> |**MEIcoder (72x128px)**|.340|.646|.997|.980|
>
> We would also like to note that we did not retrain a new encoder and generate new MEIs for the higher-resolution stimuli. Instead, we reused the MEIs generated with the encoding model trained originally on the setting with 36 x 64 px images. Encoder retraining and MEI generation tailored for this higher resolution could potentially bring additional improvements. We will add these results, including visual comparison, which we cannot share now due to NeurIPS policies, along with the interpretability results from M1, to the final manuscript.
>
> Regarding the extension to **higher visual areas**, we want to clarify that our focus on V1 is a deliberate and principled choice central to the paper’s primary motivations. As stated in our manuscript, our goals are to understand the nature of early visual representations through decoding and to advance neuroengineering applications like visual prosthetics. Both domains are primarily concerned with the precise geometric structure of visual stimuli as encoded in V1, rather than the abstract semantic content represented in higher cortical areas. Rigorous experimentation on data from higher visual areas and possible modifications of MEIcoder are unfortunately infeasible due to the short time available for this rebuttal and computational constraints, but represent an exciting avenue for future work.
>
> ---
>
> **m1:** To address the question of MEIcoder’s sensitivity to MEI quality, we re-ran training and evaluation on the Brainreader dataset with MEIs with varying levels (standard deviation, std) of Gaussian noise. As shown in the table below, the performance degrades relatively slowly. In fact, even with highly noisy MEIs (std=1 and std=3 for MEIs with initial pixel values between -1 and 1), the performance remains on par or better than that of the baselines (Table 1).
>
> ||SSIM|PixCorr|Alex(2)|Alex(5)|
> |-|-|-|-|-|
> |**MEIcoder (no noise)**|.400|.679|.998|.990|
> |**MEIcoder (std=0.2)**|.402|.675|.997|.982|
> |**MEIcoder (std=0.5)**|.390|.670|.990|.938|
> |**MEIcoder (std=1)**|.332|.625|.990|.937|
> |**MEIcoder (std=3)**|.287|.568|.984|.943|
>
> ---
>
> **m2:** Our hypothesis is that MEIcoder’s advantage is most pronounced on datasets with higher levels of intrinsic neural noise. This is supported by our finding that the trained encoder's MSE is the highest on the SENSORIUM dataset, on which MEIcoder also achieves the largest improvements over baselines (Tables 1 and 2). Our hypothesis is that the MEI prior and the adversarial objective make our decoder more robust to this noise. The MEI provides a stable template of what the neuron prefers to see, while the adversarial loss encourages the generation of sharp, plausible image structures even when the neural signal is ambiguous.
>
> ---
>
> **m3:** While the entire MEI generation, including encoder training, took around 70 minutes (15 min training + 55 min generation), the decoder training required 11 hours on one NVIDIA Tesla V100 GPU. This shows that the computational overhead of MEI generation is small relative to the decoder training itself. With regard to parameter efficiency, our claim that MEIcoder is parameter-efficient lies in the fact that we:
> 1. Use a CNN architecture, which achieves efficiency by parameter-sharing and by allowing only local connections.
> 2. Reuse the core module (backbone) across datasets from different subjects (e.g., different mice) and only train new readin modules. By sharing the core module, MEIcoder reduces the number of parameters threefold.
>
> ---
>
> **m4:** Our choice of SSIM was primarily motivated by its established ability to better capture perceptually relevant structural information compared to pixel-wise losses like MSE. As argued and demonstrated in the original SSIM paper and subsequent works such as [2], it aligns more closely with the human visual system’s perception of image quality and is often a suitable choice in image restoration applications [3].
>
> *[2] Hamid Sheikh, et al. (2006). A Statistical Evaluation of Recent Full Reference Image Quality Assessment Algorithms. IEEE Transactions on Image Processing (Volume: 15, Issue: 11).*
>
> *[3] Hang Zhao, et al. (2017). Loss Functions for Image Restoration With Neural Networks. IEEE Transactions on Computational Imaging (Volume: 3, Issue: 1).*
>
> ---
>
> > **Question 1:** *Which visual features are reconstructed with high fidelity?*
>
> We thank the reviewer for this insightful question. We performed this analysis and found two key patterns. First, the decoder’s performance directly reflects the known tuning properties of V1 neurons: it consistently reconstructs distinct, high-contrast features like corners and edges with high fidelity, while struggling with low-frequency information such as gradual changes in shading. This is a reflection of the V1 code, which is dominated by edge-detecting neurons that provide a sparse signal for uniform surfaces. Second, we identified a more global failure mode where images containing dense, high-frequency textures across a relatively small area of the image lead to a globally degraded reconstruction, an effect especially pronounced in the SENSORIUM 2022 dataset. This suggests that while V1 robustly encodes local details, the decoder can be “overwhelmed” when trying to synthesize a coherent global percept from a highly complex population signal, hinting at why the brain requires hierarchical processing. Together, these and further analyses provide valuable insights into both the encoding capacity and integrative challenges of the V1 population code, and we will add a dedicated section with illustrative examples to the revised manuscript.

---

> ### Comment · Reviewer_KLSf · 2025-08-03
>
> Thank you for the thoughtful and substantial rebuttal. I appreciate the additions, especially the new concept-based analysis using NMF and the exploration of visual features driving neuron responses. These are convincing and address my main concern (M1) regarding scientific insight. I agree that this direction frames MEIcoder not just as a decoder, but as a tool to better understand V1 computation and I hope to see them in the main paper.
>
> Regarding your new concept-based analysis (M1 and Q2), I would like to point out that similar techniques already exist in the interpretability literature. The combination of NMF with sensitivity analysis is really-closely related to the CRAFT method [1], which use NMF + Sensitivity analysis to derive importance. Other methods such as ACE[2], ICE[3], and more recently Sparse Autoencoder (SAE)[4,5] have also explored these ideas. It would strengthen the final version to position your analysis in relation to them.
>
> Aside from that, I found the rebuttal honest and precise. I will increase my score to 5, and I now feel more confident in the contribution. Good luck with the final decision.
>
> [1] CRAFT: Concept Recursive Activation FacTorization for Explainability, Fel & al.
>
> [2] Towards Automatic Concept-based Explanations, Ghorbani & al.
>
> [3] Invertible Concept-based Explanations for CNN Models with Non-negative Concept Activation Vectors, Zhang & al.
>
> [4] Archetypal SAE: Adaptive and Stable Dictionary Learning for Concept Extraction in Large Vision Models, Fel & al.
>
> [5] Steering CLIP’s vision transformer with sparse autoencoders, Joseph & al.

---

> > ### Author Response · Authors · 2025-08-06
> >
> > We thank the reviewer for their positive feedback, for increasing their score, and for providing these highly relevant pointers to the interpretability literature. We agree that our analysis shares methodological tools with methods like CRAFT [1], particularly the use of NMF and sensitivity analysis. The main difference between CRAFT and our analysis is that we employ sensitivity analysis to derive the importance of inputs (neuronal responses) on intermediate features (coefficients for combining NMF concepts). CRAFT, on the other hand, uses sensitivity analysis to derive the importance of intermediate features (coefficients for combining NMF concepts) on the output of the network. To measure the importance of inputs on the intermediate features, CRAFT uses implicit differentiation and gradient-based attribution maps.
> >
> > As the reviewer suggested, we will add our concept-based analysis and the exploration of visual features to the paper, and include a discussion that positions our work relative to CRAFT, ACE [2], ICE [3], and Sparse Autoencoders [4,5]. We are grateful for this constructive dialogue, which has helped strengthen the paper and has opened up interesting avenues for future work.
> >
> > *[1] CRAFT: Concept Recursive Activation FacTorization for Explainability, Fel & al.*
> >
> > *[2] Towards Automatic Concept-based Explanations, Ghorbani & al.*
> >
> > *[3] Invertible Concept-based Explanations for CNN Models with Non-negative Concept Activation Vectors, Zhang & al.*
> >
> > *[4] Archetypal SAE: Adaptive and Stable Dictionary Learning for Concept Extraction in Large Vision Models, Fel & al.*
> >
> > *[5] Steering CLIP’s vision transformer with sparse autoencoders, Joseph & al.*

---

### Official Review · Reviewer_1gGm · 2025-06-30

**Clarity:** 4
**Significance:** 3
**Originality:** 3
**Rating:** 5
**Confidence:** 5

**Summary:**

In this work, the authors introduce MEIcoder, a method for decoding the image presented to a sensory neural population based on their responses. The MEIcoder is unique in that it combines the use of MEIs as the characteristic features of the neuron, a structural similarity index measure to assess loss in training the decoder, and an adversarial training approach to refine the decoded image. MEIcoder outperforms other recent decoding approaches based on numerous metrics of decoded image quality.

**Questions:**

* As the authors noted, the MEI can be seen as a nonlinear counterpart to the traditional linear receptive field (with the caveat that MEI often fails to serve as an effective filter for a linear-nonlinear model of the neuron). I’m curious to learn how much of the result would hold if the MEIs were replaced by the simple gradient-based linear receptive field of each neuron?
* The authors present little to no analyses on the trained decoder’s neuron embeddings and the context representations $C$. What information can be obtained about the neurons based on these fitted quantities?
* Generally, I would have liked to see more discussion on how this decoding technique may be used to understand the properties of the neurons being decoded. In particular, what the fitted parameters in the decoder may be able to tell about the nature of the neuron and how this may change as this technique is applied to more nonlinear, higher-order sensory areas?

**Ethical Concerns:**

["NO or VERY MINOR ethics concerns only"]

**Final Justification:**

I maintain my score.

**Limitations:**

Yes

**Paper Formatting Concerns:**

None noted

**Quality:**

4

**Strengths And Weaknesses:**

Quality:
* The authors offer very well-written, to-the-point, and helpful introduction to the subject matter
* The authors provide a comprehensive review of the related work and successfully compare their work to the previous work, often directly through replication of the published method. This significantly strengthens the authors’ claims.
* All technical and mathematical expressions are coherent and accurate

Clarity:
* The paper is very well structured and clearly written, with an excellent and to-the-point summary of the main contributions of the work.
* The term FT is used without definition. While I infer this stands for “feature transfer”, a concrete definition in the text would be useful.

Significance:
As the author notes, there has been a significant advancement in employing recent advances in ML to develop state-of-the-art encoding / system-identification models, and it is precisely the advent of these models that has led to the development of neuron characterizing techniques such as MEI. On the other hand, achieving effective decoding of sensory stimuli presented based on a sensory neural population has remained challenging, especially in the regime of limited neuron counts. While most recent efforts in utilizing generative AI, especially those based on diffusion models, have seen success, as the authors have noted, these methods are prone to “hallucination”.
* The MEIcoder presented in this work offers a simple yet effective method to “decode” sensory inputs from a neural population, employing MEI as the effective template that each neuron encodes about the image. I find this use of MEI quite innovative and exciting. However, I would have liked to see analyses showing that MEI is necessary, especially in comparison to more traditional neural characterizations such as linear receptive field.
* Also, from the analyses presented in the work, it remains unclear how much the success of this method depends on the linearity (or for that matter, simple linear-nonlinear nature) of the neurons. In other words, would this kind of MEI-based decoding prove effective for highly nonlinear neurons?

Originality:
* Use of MEIs as a strong characterization basis in decoding neural responses back to the input image is a very novel application of MEIs, which have been developed and used primarily as a method of verifying the fidelity of encoding models that were used to generate the MEIs and to qualitatively describe the nature of the neuron.
* The use of context representations to modulate the contributions of the individual MEI is highly innovative, and understanding the context representation $C$ in itself could lead to potential insights into exactly how each neuron contributes to the scene understanding.

---

> ### Author Rebuttal · Authors · 2025-07-31
>
> We thank the reviewer for their positive and insightful review. We are delighted that the reviewer recognized the novelty of our work, particularly the “innovative and exciting” use of MEIs as a decoding prior and the originality of our context modulation mechanism. We also sincerely appreciate the reviewer’s acknowledgement of the paper’s clarity of writing and of the successful comparison to previous work, which significantly strengthens our claims. In our response, we address each reviewer’s concern individually.
>
> ---
>
> > **Clarity:** *The term FT is used without definition.*
>
> We thank the reviewer for pointing out this lack of clarity. We will explicitly define “FT” as “fine-tuning” upon its first use in the revised manuscript (Section 4.4, line 234) to avoid any confusion.
>
> ---
>
> > **Significance, Question 1:** *Necessity of MEIs and comparison to gradient-based linear receptive fields of neurons.*
>
> We appreciate this insightful question. To directly test the necessity of the MEI prior, we included in the initial submission an ablation study where we removed MEIs from the readin module and used only the context representations as the initial reconstruction template. As can be seen in Figure 4, this led to a significant degradation of performance. With regards to the comparison of MEIs to more traditional neural characterizations, we conducted additional experiments where we replaced MEIs with gradient-based linear receptive fields (LRFs) obtained from regularized regression trained on neural responses from the Brainreader dataset. As shown in the table below, this also leads to degraded performance, but not as severe as with a complete removal of neural characterization as done in the initial ablation study. This demonstrates the importance of biologically informed computational prior, MEIs in particular, for the decoder’s performance.
>
> ||SSIM|PixCorr|Alex(2)|Alex(5)|
> |-|-|-|-|-|
> |**MEIcoder (original)**|.400|.679|.998|.990|
> |**MEIcoder (LRFs)**|.364 (-9%)|.663 (-2.4%)|.998 (-0%)|.949 (-4.1%)|
>
> ---
>
> > **Significance:** *Would MEI-based decoding prove effective for highly nonlinear neurons?*
>
> This is indeed a very interesting question. To answer it most thoroughly, one would repeat our experiments in higher visual areas. Unfortunately, in the very limited time available for this rebuttal, we were not able to run such complex experiments on new data. Nevertheless, we have attempted to address the question at least within the context of the V1 data that we have available. We have calculated a so-called non-linearity index (NLI), which estimates how non-linear individual neurons are [1]. It is calculated as the ratio between the prediction power of a linear encoding model fitted to the data and the prediction power of a state-of-the-art non-linear encoding model fitted to the data. As can be seen in the table below, our ability to decode images from the more non-linear neurons is very similar to our ability to decode from more linear neurons, suggesting that our method can handle non-linearity of the code very well, at least within the context of V1.
>
> ||SSIM|PixCorr|Alex(2)|Alex(5)|
> |-|-|-|-|-|
> |**3000 most nonlinear neurons**|.321|.630|.999|.958|
> |**3000 least nonlinear neurons**|.324|.653|.996|.939|
> |**3000 least nonlinear neurons (nonzero NLI)**|.345|.663|.999|.979|
>
> *[1] Ján Antolík, et al. (2016). Model Constrained by Visual Hierarchy Improves Prediction of Neural Responses to Natural Scenes. PLoS Computational Biology, 12, 1–22.*
>
> ---
>
> > **Originality, Question 2, Question 3:** *Potential insights into how each neuron contributes to the scene understanding. What information can we obtain about the neurons based on fitted quantities such as neuron embeddings, context representations, and other decoders’ parameters? How may this change as this technique is applied to more nonlinear, higher-order sensory areas?*
>
> We thank the reviewer for highlighting the innovative nature of our context modulation and for these suggestions on exploring the model’s interpretability. We have performed a new set of analyses to directly probe the function of the learned parameters and provide insights into the decoding process.
>
> **Insights into neuronal contributions:** We implemented a concept-based analysis that combines two techniques: First, we use Non-negative Matrix Factorization (NMF) to learn a dictionary of 32 “visual concepts” (or basis features) from the feature maps at different layers of our decoder. Second, we perform a sensitivity analysis to observe how systematically manipulating the response of a single neuron activates these specific concepts. This provides a direct, interpretable link from a single biological neuron’s activity to the activation of learned hierarchical visual features within the decoder. Our key findings from this analysis include:
> 1. *Hierarchical feature learning:* We found that for the majority of neurons, the bright regions of the visual concepts they most strongly activate become smaller and more focused as we traverse through the decoder's layers. This suggests the model learns a hierarchy, starting with coarse, broad features and progressively refining them into more detailed structures, with the final location remaining consistent with the neuron’s original MEI.
> 2. *Shared and specialized neuron functions:* Many neurons’ top-activated concepts include a feature that encodes the brightness of the border around the whole image. This suggests that many neurons encode global lighting conditions, while concurrently being individually specialized in encoding fine-grained local structures at different places in the visual field (as supported by MEIs).  This might hint at how, through lateral cortical processing, our cortex fills in information where it is missing (like in the blind spot, where no photoreceptors are present).
> 3. *Dynamic object representation:* We identified a few neurons for which incrementally increasing their response results in an incremental shift of a dark object in the reconstructed image. This highlights that the model has learned not just to represent static features, but also to dynamically alter the scene’s composition based on the activity of specific, influential neurons. Interestingly, this emergent property of our model mirrors key findings about functional asymmetries in the visual cortex. More specifically, a study in macaque V1 by [2] found a significant over-representation of “black-dominant” (OFF) neurons in the corticocortical output layers 2/3. As the authors noted, their results strongly suggested that black-over-white human perceptual preference is generated or greatly amplified in V1.
>
> These new analyses begin to leverage MEIcoder as a tool for scientific discovery, revealing how it has learned to interpret and combine the contributions of individual neurons to form a coherent reconstruction, demonstrating its utility for studying visual neural codes. We have prepared visualizations of these findings, and while NeurIPS policy prohibits ways of sharing them during the rebuttal period, we will add a dedicated section with these figures and a detailed discussion in the appendix of the final manuscript.
>
> **Neuron embeddings & context representations:** We investigated whether neuron embeddings implicitly learn the retinotopic organization of V1. Using Multidimensional Scaling (MDS) to project the 32-dimensional neuron embeddings into a 2D space, we did not find that they would recover the spatial coordinates of the neurons as learned in the Gaussian readout module of our encoder architecture [3]. Regarding the context representations, our sensitivity analysis mentioned above provides a direct visualization of their function. We found that their response-dependent spatial gain map, which multiplies the base MEIs, is often spread over the whole MEI and sometimes highlights a small part of the MEI while inhibiting other parts. We acknowledge that applying other interpretability techniques and relating the findings from MDS and related dimensionality-reduction techniques to known neuronal properties is an interesting avenue for future work.
>
> **Nonlinear, higher-order sensory areas:** Our aforementioned analysis with the non-linearity index (NLI) showed that MEIcoder is robust to the nonlinearities present within V1, but we agree that areas like V4 or IT might present different and interesting challenges. We hypothesize that the principles of our analysis would remain, but the findings would shift. The NMF concepts would likely evolve from focused local structures and Gabor-like edges to more holistic object parts or textures. This represents an exciting direction for future research, where interpretability frameworks could be used to map the functional organization of these higher-order areas.
>
> *[2] Chun-I Yeh, et al. (2009). “Black” responses dominate macaque primary visual cortex v1. The Journal of neuroscience: the official journal of the Society for Neuroscience, 29(38), 11753–11760.*
>
> *[3] Konstantin-Klemens Lurz, et al. (2021). Generalization in data-driven models of primary visual cortex. bioRxiv, doi: 10.1101/2020.10.05.326256.*

---

> > ### Comment · Reviewer_1gGm · 2025-08-06
> >
> > I thank the authors for the details and thoughtful responses to my comments and questions. It was nice to see the additional analyses conducted in comparing to more traditional linear methods and the additional assessment performed on the nonlinearity further highlights the high applicability of this methodology to higher order brain areas. As I have already had it marked for solid Accept at 5, I will maintain the score.

---

### Official Review · Reviewer_2fvu · 2025-07-01

**Clarity:** 3
**Significance:** 3
**Originality:** 3
**Rating:** 5
**Confidence:** 4

**Summary:**

In order to reconstruct high-quality visual stimuli at the pixel level from limited neural data, this work introduces MEIcoder, which incorporates neuron-specific most exciting inputs into a visual-stimuli decoding framework. In the experiments on three neural datasets, MEIcoder achieves the best reconstruction performance compared to alternative models. Furthermore, in scaling experiments, the results demonstrate the superiority of MEIcoder in dealing with small numbers of neurons and data points.

**Questions:**

1. When the number of neurons and the resolution of visual stimuli increase, is a more complex encoding model required to calculate MEIs, and can the computational cost still be ignored?
2. Are there repeated trials for each visual stimulus in each dataset? How are the training set, validation set, and test set divided, and do they contain repeated images? Can the authors provide a more detailed description of the visual stimuli used in each dataset?

**Ethical Concerns:**

["NO or VERY MINOR ethics concerns only"]

**Final Justification:**

Based on the additional results provided by the author, I raise my score.

**Limitations:**

yes

**Quality:**

3

**Strengths And Weaknesses:**

Strengths:

1. This work is well-motivated, and introducing MEIs into decoding models is novel and meaningful.
2. MEIcoder outperforms several outstanding and well-known models in multiple metrics and visualization, demonstrating its effectiveness.
3. Comprehensive ablation studies prove the role of each module of MEIcoder, especially MEIs.

Weaknesses:

1. My primary concern is the fairness of the experiments. First, the authors do not provide results of the datasets used in baseline models and instead conduct comparisons on new datasets (where only the Brainreader dataset has been used in InvEnc). Second, MindEye2 is more focused on semantically reconstructing visual stimuli, which inherently puts it at a disadvantage in low-level or even pixel-level metrics. Could the authors provide results using higher-level metrics employed in MindEye2? Finally, [1] has reconstructed high-quality movie stimuli from the neural activity of hundreds of mouse visual cortex neurons. I think this study should be included in the comparison, as it also targets small datasets and pixel-level reconstruction.
2. Since the visual stimuli in this work have low resolution, it is unclear whether the model can be scaled up to handle higher-resolution visual stimulus reconstruction.

[1] YeChen, et al. Decoding dynamic visual scenes across the brain hierarchy. PLOS Computational Biology 2024.

---

> ### Author Rebuttal · Authors · 2025-07-31
>
> We thank the reviewer for their thoughtful review. We sincerely appreciate the reviewer's recognition of the originality of the method by introducing MEIs into the image reconstruction process. We are also grateful for the reviewer's acknowledgment that MEIcoder outperforms several well-known baselines on multiple different datasets on both quantitative and qualitative levels, and that our extensive ablations strongly support our design decisions.
>
> ---
>
> > **Weakness 1**: *Fairness of experiments.*
>
> The reviewer raises important points about the fairness and comprehensiveness of our experimental evaluation. We address each of the concerns below.
>
> **On the choice of datasets:** We agree that comparing on the original datasets of baseline methods is ideal, which is also why we chose one of our two biological datasets to come from the baseline method InvEnc. However, for MindEye2, the original data is from functional magnetic resonance imaging (fMRI), which does not allow identifying detailed receptive fields or MEIs, essential for our method, and in turn decoding of fine spatio-temporal details of target images. This makes this dataset less suitable for testing our method, which focuses on decoding the exact fine geometric details, which are important for studying neural codes in early visual areas and for applications such as the development of prostheses for those with vision impairment. For MonkeySee, the dataset has not been released together with the published paper, and our repeated email inquiries to the authors were not answered. Similarly, data used by EGG was not released with the paper, and we were unable to find it elsewhere on the internet. This state of affairs also motivated us to introduce an easily accessible benchmark data corpus of over 160,000 data points, which we hope will make the area of brain activity decoding more accessible.
>
> **On the evaluation metrics:** While we wish the emphasize that the primary motivation of our work is to develop a method focused on high-fidelity reconstruction of low-level geometric structures in images (i.e. decoding the fine geometric detail of target images rather than just their overall semantics), which is crucial for studying visual representations in early visual areas (such as V1 or V2), as well as applications such as visual prosthetics targeting these low-level visual areas, we agree that a broader comparison is valuable. To this end, using the implementation from MindEye2 [2], we have evaluated all methods using the two-way identification accuracy with InceptionV3 (“Incep”) and CLIP (ViT-L/14) embeddings, as well as the average correlation distance in the embedding space of EfficientNet-B1 (“Eff”) and SwAV-ResNet50 (“SwAV”). As can be seen in the tables below, our method remains highly competitive even on these semantic metrics (Incep and CLIP: the higher the better, Eff and SwAV: the lower the better). On both the Brainreader and SENSORIUM 2022 datasets, MEIcoder achieves the best performance, while on the Synthetic Cat V1 data, it outperforms the other baselines on two out of four metrics, remaining on par on the other two performance measures. We can also see that the MEIcoder fine-tuned (“FT”) from multi-subject to single-subject data performs the best on the biological datasets. This further demonstrates MEIcoder’s ability to generalize across data from different subjects.
>
> - **Brainreader:**
> ||Incep|CLIP|Eff|SwAV|
> |-|-|-|-|-|
> |**InvEnc**|.627|.636|.606|.448|
> |**EGG**|.749|.610|.652|.497|
> |**MonkeySee**|.655|.547|.653|.486|
> |**MindEye2**|.772|.636|.571|.415|
> |**MindEye2 (FT)**|.725|.652|.584|.465|
> |**MEIcoder**|.799|.679|.586|.489|
> |**MEIcoder (FT)**|**.817**|**.702**|**.520**|**.408**|
>
> - **SENSORIUM 2022:**
> ||Incep|CLIP|Eff|SwAV|
> |-|-|-|-|-|
> |**InvEnc**|.600|.564|.492|.243|
> |**EGG**|.673|.579|.449|.232|
> |**MonkeySee**|.614|.513|.551|.301|
> |**MindEye2**|.643|.590|.440|.235|
> |**MindEye2 (FT)**|.650|.595|.443|.219|
> |**MEIcoder**|.727|.590|.440|.276|
> |**MEIcoder (FT)**|**.746**|**.620**|**.419**|**.216**|
>
> - **Synthetic Cat V1:**
> ||Incep|CLIP|Eff|SwAV|
> |-|-|-|-|-|
> |**InvEnc**|.884|**.827**|.326|.196|
> |**EGG**|.766|.675|.283|.214|
> |**MonkeySee**|.796|.760|**.272**|.193|
> |**MindEye2**|.798|.780|.274|.208|
> |**MEIcoder**|**.898**|.761|.277|**.181**|
>
> *[2] Paul S. Scotti, et al. (2024). MindEye2: Shared-Subject Models Enable fMRI-To-Image With 1 Hour of Data. ICLR 2024.*
>
> **On including [1] as a baseline:** We appreciate the reviewer for pointing us to this highly relevant work. Using the original codebase of [1], we evaluated it on our benchmark datasets. We find that while it performs strongly, MEIcoder still demonstrates superior performance in the majority of cases:
> - **Brainreader:**
> ||SSIM|PixCorr|Alex(2)|Alex(5)|
> |-|-|-|-|-|
> |**MEIcoder**|**.400**|**.679**|**.998**|**.990**|
> |**Chen 2024 [1]**|.256|.638|.930|.730|
>
> - **SENSORIUM 2022:**
> ||SSIM|PixCorr|Alex(2)|Alex(5)|
> |-|-|-|-|-|
> |**MEIcoder**|**.331**|.503|**.988**|**.896**|
> |**Chen 2024 [1]**|.287|**.539**|.656|.649|
>
> - **Synthetic Cat V1:**
> ||SSIM|PixCorr|Alex(2)|Alex(5)|
> |-|-|-|-|-|
> |**MEIcoder**|**.774**|.777|**.994**|**.987**|
> |**Chen 2024 [1]**|.637|**.792**|.927|.776|
>
> We will include these comparisons and discussions in the revised manuscript. Again, thank you for this excellent suggestion.
>
> *[1] Ye Chen, et al. Decoding dynamic visual scenes across the brain hierarchy. PLOS Computational Biology 2024.*
>
> ---
>
> > **Weakness 2:** *Use of low-resolution images.*
>
> We agree about the importance of demonstrating scalability to higher resolutions. To address this concern, we have conducted a new experiment on the Brainreader dataset using a two-times higher image resolution of 72x128 pixels. We are pleased to report in the table below that MEIcoder’s performance is maintained at this higher resolution, even with very limited hyperparameter search due to time constraints. In fact, although the task of reconstructing larger images is inherently more difficult, the higher-resolution MEIcoder still outperforms the best alternative baseline from the low-resolution setting in Table 1 of our paper.
>
> ||SSIM|PixCorr|Alex(2)|Alex(5)|
> |-|-|-|-|-|
> |**MEIcoder (36x64px)**|.400|.679|.998|.990|
> |**MEIcoder (72x128px)**|.340|.646|.997|.980|
>
> This demonstrates that our approach is not limited to low-resolution stimuli and can effectively scale to more detailed visual inputs. While this format does not allow posting images, and NeurIPS’ rules prohibit sharing (anonymous) links, we can confidently say that the higher-resolution image reconstructions look sharp and capture fine-grained structural details very well. It is also important to note that due to the time constraints, we did not train a more complex encoding model and generate new MEIs; instead, we reused the MEIs generated with the encoder trained on the lower resolution setting. Retraining of the encoder and the generation of MEIs tailored for this higher resolution could potentially bring further improvements. We will add these new results, including a comparison with baselines and a discussion on scalability, to the final manuscript.
>
> ---
>
> > **Question 1:** *A more complex encoding model and its computational cost.*
>
> This is an excellent question regarding the computational trade-offs of our framework. For the number of neurons, we did not need to increase the complexity of the encoding model when we scaled up from 8k neurons (Brainreader and SENSORIUM 2022 datasets) to 46k neurons (Synthetic Cat V1 dataset). The architecture and most of the hyperparameters were kept the same. With regard to the resolution of visual stimuli, we also did not need to increase the complexity of the encoding model to arrive at the results from training and evaluating MEIcoder on higher-resolution images. In fact, we reused the MEIs generated for the lower-resolution images. This shows that we can keep the computational cost of encoder training and MEI generation the same when we scale to higher-resolution images.
>
> In summary, we can state that the computational cost of encoder training and MEI generation does not constrain the application of our method to larger-scale problems.
>
> ---
>
> > **Question 2:** *Details about data.*
>
> We thank the reviewer for requesting further clarification on our experimental setup and apologize for any ambiguity in the original submission.
>
> **Repeated trials:** To clarify, repeated trials were only available for the test sets, which is a common practice in neuroscience to obtain a more stable, averaged response for evaluation. Specifically, there are 40 trials for Brainreader, 10 for SENSORIUM 2022, and 100 for Synthetic Cat V1. All data points in the training and validation sets were from single trials.
>
> **Data splits and duplicates:** For Brainreader, we used the original train/validation/test splits provided by the dataset’s authors. For SENSORIUM 2022, we randomly subdivided the original training data into our own training (4,500 points) and validation (484 points) sets; a separate test set was provided by the organizers of the SENSORIUM 2022 competition. For Synthetic Cat V1, we partitioned a large set of generated data into training (45,000), validation (5,000), and test (250) sets. We have verified that there is no overlap of images across the splits within each dataset, nor across the training and validation splits of different datasets. The only duplicate images are in the test sets of Brainreader and SENSORIUM 2022, where the original authors used the same ImageNet images.
>
> **Stimuli description:** All images, except hand-selected ones in the test sets of Brainreader and SENSORIUM 2022, were randomly sampled from ImageNet. The final sizes of the grayscale-mapped stimuli were: 36 x 64 px (Brainreader), 22 x 36 px (SENSORIUM 2022), and 20 x 20 px (Synthetic Cat V1).
>
> We will add all the above information to the final manuscript.

---

> > ### Comment · Reviewer_2fvu · 2025-08-06
> >
> > Thanks to the author for conducting additional experiments and comparisons, which have largely addressed my concerns. I will raise my score.

---

### Decision · Program_Chairs · 2025-09-17

**Decision:**

Accept (poster)

**Comment:**

This paper introduces MEIcoder, a method for reconstructing visual stimuli from neural population activity by leveraging neuron-specific Most Exciting Inputs (MEIs), structural similarity (SSIM)-based loss, and adversarial training. The model demonstrates strong performance in low-data and low-neuron regimes, offering both technical advances and potential utility for neuroscience and brain–machine interface applications.

The reviewers highlighted several strengths: the novelty of using MEIs as biologically grounded priors, strong empirical performance across datasets, data efficiency in scarce regimes, and a well-executed evaluation including ablations and scaling experiments. Weaknesses raised included fairness of baselines and evaluation metrics, limited discussion of scalability to higher-resolution or higher-order brain areas, and the need for deeper analysis of the neuroscience insights gained from the reconstructions.

The authors’ rebuttal and follow-up experiments convincingly addressed these concerns. They added comparisons with previously omitted baselines, demonstrated MEIcoder’s robustness on semantic metrics and at higher resolutions, and provided new analyses (e.g., concept-based decomposition, sensitivity analyses, and linear RF ablations) that underscored MEIs’ necessity and interpretability. These additions strengthened the claims and reassured reviewers about both methodological rigor and broader relevance.

After discussion, consensus among reviewers was broadly positive. Initial concerns about fairness, scalability, and interpretability were largely alleviated, with reviewers explicitly raising their scores. Some lingering limitations remain regarding generalization to higher-order visual areas and broader neuroscience insights, but reviewers agreed that the work is solid, innovative, and timely. The overall sentiment converged on acceptance.

Given all this, the AC recommends the paper to be accepted, as it makes a clear and well-supported contribution to neural decoding and will likely stimulate further work at the interface of AI and neuroscience.